

# Quantum gravity, renormalizability and diffeomorphism invariance

**Tim R. Morris**

STAG Research Centre & Department of Physics and Astronomy,
University of Southampton, Highfield, Southampton, SO17 1BJ, U.K.

⋆ T.R.Morris@soton.ac.uk

## Abstract

We show that the Wilsonian renormalization group (RG) provides a natural regularisation of the Quantum Master Equation such that to first order the BRST algebra closes on local functionals spanned by the eigenoperators with constant couplings. We then apply this to quantum gravity. Around the Gaussian fixed point, RG properties of the conformal factor of the metric allow the construction of a Hilbert space $\mathfrak{L}$ of renormalizable interactions, non-perturbative in $\hbar$, and involving arbitrarily high powers of the gravitational fluctuations. We show that diffeomorphism invariance is violated for interactions that lie inside $\mathfrak{L}$, in the sense that only a trivial quantum BRST cohomology exists for interactions at first order in the couplings. However by taking a limit to the boundary of $\mathfrak{L}$, the couplings can be constrained to recover Newton's constant, and standard realisations of diffeomorphism invariance, whilst retaining renormalizability. The limits are sufficiently flexible to allow this also at higher orders. This leaves open a number of questions that should find their answer at second order. We develop much of the framework that will allow these calculations to be performed.



# 1   Introduction and review

As is well known, quantum gravity suffers from the problem that it is not renormalizable when perturbatively expanded in $\kappa = \sqrt{32\pi G}$ (where $G$ is Newton's gravitational constant) and Planck's constant $\hbar$ [1–4]. By using the Wilsonian renormalization group (RG), we discovered a promising route out of this impasse, which points to a theory that would still be perturbative in $\kappa$ but is now non-perturbative in $\hbar$ [5, 6].

The key is the so-called conformal factor instability which has up to now generally been side-stepped, or dealt with by analytically continuing the conformal factor functional integral along the imaginary axis: $\varphi \mapsto i\varphi$ [7]. If we do not continue $\varphi$ to imaginary values, a sensible (in particular Banach) space of interactions around the Gaussian fixed point is only achieved if a quantisation condition is imposed on bare interactions involving $\varphi$, which is that they should be square integrable over amplitude $\varphi \in (-\infty, \infty)$ with weight

$$\exp\left(\varphi^2/2\Omega_\Lambda\right), \tag{1.1}$$

where $\Omega_\Lambda = |\langle\varphi(x)\varphi(x)\rangle|$ is the (magnitude of the) free propagator at coincident points, regularised by an UV (ultraviolet) cutoff $\Lambda$. The result is $\mathfrak{L}_-$, a Hilbert space of interactions

with a number of marvellous properties [5, 8]. For non-derivative interactions, it is spanned by a tower of operators

$$\delta_\Lambda^{(n)}(\varphi) := \frac{\partial^n}{\partial \varphi^n} \delta_\Lambda^{(0)}(\varphi), \qquad \text{where} \qquad \delta_\Lambda^{(0)}(\varphi) := \frac{1}{\sqrt{2\pi\Omega_\Lambda}} \exp\left(-\frac{\varphi^2}{2\Omega_\Lambda}\right) \qquad (1.2)$$

(integer $n \geq 0$), all of which are relevant at the linearised level, their dimensions being $[\delta_n] = -1 - n$ (in four dimensional spacetime). Since $\Omega_\Lambda \propto \hbar$, they are non-perturbative in $\hbar$. To get interactions for general gravitational fluctuations we need also the traceless fluctuation field $h_{\mu\nu}$ whose self-interactions form a Hilbert space in the usual way in quantum field theory, *i.e.* spanned by polynomials. Together we then get the full Hilbert space of $\mathfrak{L}$ of bare interactions [5] that are square integrable under

$$\exp \frac{\varphi^2 - h_{\mu\nu}^2}{2\Omega_\Lambda} \qquad (1.3)$$

(in four dimensional spacetime), and which is spanned by operators whose top term has the form

$$\delta_\Lambda^{(n)}(\varphi)\, \sigma(\partial, \partial\varphi, h), \qquad (1.4)$$

where $\sigma(\partial, \partial\varphi, h)$ is a local Lorentz invariant monomial involving some or all of the components indicated (and thus $h_{\mu\nu}$ can appear here differentiated or undifferentiated or not at all). These operators have scaling dimension $D_\sigma = d_\sigma + [\delta_n]$, where $d_\sigma = [\sigma(\partial, \partial\varphi, h)]$ is the engineering dimension of $\sigma$. From the Wilsonian RG, we know that if we can build the theory using only bare operators that are (marginally) relevant, implying $D_\sigma \leq d$ (where $d$ is the spacetime dimension) and corresponding couplings $[g_n^\sigma] \geq 0$, then we can build a continuum limit [9, 10].[1] Since this continuum limit is constructed at the Gaussian fixed point, it corresponds to a perturbatively renormalizable quantum field theory.

For every $\sigma$, infinitely many of the operators (1.4) are relevant. Since the perturbative expansion of the classical action in $\kappa$, leads to interactions of arbitrarily high power in the fluctuation field, this quantisation would seem to be tailor-made for finding a route out of the impasse. A crucial question however is whether, on restriction to the space spanned by the (marginally) relevant operators, it is possible to build in a quantum version of diffeomorphism invariance at the renormalized level, and thus obtain a perturbatively renormalizable theory of quantum gravity. The answer appears to be yes: diffeomorphism invariance is implemented by taking suitable limits to the boundary of $\mathfrak{L}$, thus enlarging the Hilbert space with states of infinite norm whilst nevertheless staying renormalizable, while at the same time constraining couplings to give $\kappa$ in appropriate circumstances.

In ref. [5], it was already shown that there is no solution within $\mathfrak{L}$ to incorporating diffeomorphism invariance if we ask this question of a classical action, where the problem reduces to choosing a parametrisation of the metric. However such a question presupposes the existence of the corresponding classical limit. Since the operators (1.2) on which the theory is to be built, all vanish as $\hbar \to 0$, there is no such correspondence principle.

The correct question to ask is whether BRST invariance [11–14] can be consistently implemented directly at the quantum level [5]. At its most general, we can ask whether there is any such non-trivial BRST algebra in $\mathfrak{L}$ that smoothly goes over to linearised diffeomorphism invariance in the $\kappa \to 0$ limit. By non-trivial we mean in particular that there should exist

---

[1]Actually we can also include irrelevant couplings as discussed for this case in ref. [5], but they would have to vanish fast enough as $\Lambda \to \infty$.

quasi-local interactions[2] that are BRST closed but not exact, the latter corresponding merely to quasi-local reparametrisations of the field variables.

At the classical, and strictly local, level, this question is addressed by the well developed subject of BRST cohomology [18–22] (for reviews see [23–25]), exploiting properties of the Batalin–Vilkovisky antibracket formalism [26–29] to ask simultaneously for consistent deformations of linearised diffeomorphism invariance and local actions that would realise it, whilst automatically taking into account all local reparametrisations. Under rather broad assumptions the answer is unique: General Relativity is the only solution [22]. (For earlier alternative approaches, see refs. [30–39].)

What we require however is the generalisation of this question to the quantum case, appropriately regularised in a way that respects the Wilsonian RG properties of the operators. Thus first we need a consistent combination of the Wilsonian RG and Batalin–Vilkovisky framework.

This has been addressed in refs. [40–48], and adapted and applied especially to QED and Yang-Mills theory.[3] However as reviewed in sec. 2.5 these formulations have drawbacks, because they either leave the Batalin–Vilkovisky measure term $\Delta$ [26, 27] unregularised when acting on local functionals, or destroy the locality of the BRST (and Koszul-Tate) transformations. This in turn would imply that the quantum BRST algebra cannot close on local functionals. It would thus force us to work even at first order in the 'deformation parameter' $\kappa$, with expansions to infinite order in derivatives.

In sec. 2.4, we show however that this is not inevitable. There exists a particularly natural formulation that solves these problems. As we explain in sec. 2.6, at first order in $\kappa$ it results in a regularised $\Delta$ (when acting on arbitrary local functionals), but leaves the rest of the BRST structure unmodified. It thus allows us to investigate (in sec. 7.2) its first-order closure on local functionals, and then by extension its closure in the space of quasi-local functionals. Most importantly as we show in sec. 2.6, it allows us to define the quantum BRST cohomology to lie within the space spanned by the eigenoperators, appropriately extended as we explain in sec. 5. This is therefore an especially nice formulation for the questions we want to address, and henceforward provides the starting point for full $\kappa$-perturbative calculations in this theory of quantum gravity.

In sec. 3, we specialise to quantum gravity to prepare the ground for these studies, first in gauge invariant basis in sec. 3.1 and then in gauge fixed basis in sec. 3.2. At this stage we work in $d$ dimensions, and in a general gauge. In sec. 3.4 we explain why the conformal mode instability is a physical effect, how it compares with the treatment in ref. [5], and how it is affected by choice of gauge. We discuss the changes needed to treat the fluctuation field as a density, or as in unimodular gravity, and explain carefully why the conformal mode instability is different from the common (mal)practice of treating the auxiliary field $b_\mu$ with wrong sign bilinear term. In sec. 3.5 we derive the propagators, first in general gauge $\alpha$. Although we expect all the physical results to be gauge independent, from here on we specialise to $\alpha = 2$ for which the propagators are particularly simple, in particular the traceless mode $h_{\mu\nu}$ and conformal factor do not then propagate into each other.

Sec. 4 is devoted to deriving the eigenoperators, culminating in the general form (4.28) which now includes the ghosts, auxiliary fields and antifields. This allows us to derive the Hilbert space of bare interactions in sec. 5 and discuss its interpretation and compatibility with BRST invariance. The BRST cohomology needs however to be defined for the renormalized interactions. For the sake of clarity, we specialise to $d = 4$ spacetime dimensions, from here

---

[2] By quasi-local we mean that the vertices must have a derivative expansion, corresponding to a Taylor expansion in dimensionless momenta $p^\mu/\Lambda$. This is a fundamental requirement of the Wilsonian RG in the continuum [15,16], corresponding to the existence of a sensible Kadanoff blocking [17]. It should be distinguished from (strictly) local (a.k.a. ultra-local [15,16]) vertices which are a finite sum of local monomials which thus have a maximum number of spacetime derivatives.

[3] For other related approaches see refs. [49–53].

on. Adapting ref. [5], we show how the infinite towers of relevant eigenoperators organise into coefficient functions and derive and discuss the properties of the renormalized coefficient functions, in particular the emergence and significance of the amplitude decay scale, to the extent that it is required in this paper. (Other aspects are discussed in refs. [5, 8].) In sec. 6.1 we furnish some examples, which will prove important in sec. 7.3.

Finally in sec. 7 we are able to turn to BRST cohomology, the main *raison d'être* for the paper. We start by recalling the classical BRST cohomology [22]. Then in sec. 7.2, we turn to the quantum BRST cohomology. We start by emphasising why it is important to define the space of functionals in which this is to be studied. Then we show how the classical cohomology is altered if we define the quantum cohomology in the Wilsonian framework as a perturbation series in $\hbar$ where we take the space to be spanned by polynomials in the fields *i.e.* following the standard quantisation. In the new quantisation, despite having at our disposal Wilsonian effective interactions with arbitrarily high powers of space-time derivatives, and despite having enlarged the algebra with a properly regularised version of the measure term $\Delta$, we will see that no BRST non-trivial interactions can be introduced while lieing strictly inside $\mathfrak{L}$. On the other hand, we show in sec. 7.3 that we can modify any standard parametrisation of the interactions so that they are an expansion only over (marginally) relevant operators in $\mathfrak{L}$, for all finite values of amplitude suppression scale $\Lambda_\sigma$. Then we show that by taking the limit $\Lambda_\sigma \to \infty$, full BRST invariance can be recovered. This is already sufficient to ensure that diffeomorphism invariance is correctly incorporated at first order in $\kappa$. We show that at higher orders the procedure has sufficient flexibility *a priori* to allow an order by order solution of the Quantum Master Equation (QME). In sec. 8 we discuss what further insights can be gained so far, and highlight some of the open issues. In sec. 9 we summarise and draw our conclusions.

## 2 Combining the QME, BRST cohomology and the Wilsonian RG

As sketched in the Introduction, we need to ask how diffeomorphism invariance is implemented directly at the quantum level while respecting the fact that the form of the interactions is dictated by the Wilsonian RG. The operators (1.2) depend for their existence on quadratic divergences. They are not well defined in dimensional regularisation [5]. Therefore to discuss renormalizability, employing a cutoff that breaks the gauge invariance seems so far to be essential [5].

### 2.1 Quantum Master Equation

Renormalizability in the presence of non-Abelian local symmetries can still be assured if we can show that the Zinn-Justin equation [54] for the Legendre effective action, can be satisfied. This takes into account the fact that the gauge invariance is realised at the quantum level through requiring a BRST invariance [11–14] and is flexible enough to accommodate its deformation under regularisation and renormalization. At the level that we will need it here, it is more elegant to work directly with the quantum fields where the modified BRST transformations can be written as

$$\delta\Phi^A = \epsilon\, Q\Phi^A \tag{2.1}$$

($\epsilon$ a Grassmann number). The $\Phi^A$ are the quantum fields, including ghost fields – and auxiliary fields to realise BRST invariance off-shell. For a non-Abelian symmetry, the BRST charge $Q$ depends on the fields themselves. We define this Grassmann odd derivation to act from the left. To renormalize, we need to supplement the bare action $\mathcal{S}[\Phi]$ with source terms $\Phi_A^*$ for these BRST transformations, so that the total action is

$$S = \mathcal{S}[\Phi] - (Q\Phi^A)\Phi_A^*. \tag{2.2}$$

The partition function is then simply

$$\mathcal{Z}[\Phi^*] = \int \mathcal{D}\Phi \, e^{-S} \,. \tag{2.3}$$

The $\Phi_A^*$ have opposite statistics to the $\Phi^A$ and are called antifields. We are using the compact Dewitt notation. Up to some choices of convention, we follow Batalin and Vilkovisky [26, 27], except that we implement gauge fixing by working in the gauge fixed basis [29, 55–59], and also retain the antifields. This is explained in sec. 2.7.

The gauge symmetry is successfully incorporated if the functional integral is invariant under (2.1), and this is true if and only if the QME is satisfied, namely $\mathcal{A} = 0$, where the QMF (Quantum Master Functional) is:[4]

$$\mathcal{A}[S] = \tfrac{1}{2}(S,S) - \Delta S \,. \tag{2.4}$$

Here, we are introducing the antibracket $(\,\cdot\,,\,\cdot\,)$ and the measure operator $\Delta$, which on arbitrary functionals $X$ and $Y$ take the form:

$$(X,Y) = \frac{\partial_r X}{\partial \Phi^A} \frac{\partial_l Y}{\partial \Phi_A^*} - \frac{\partial_r X}{\partial \Phi_A^*} \frac{\partial_l Y}{\partial \Phi^A} \qquad \text{and} \qquad \Delta X = (-)^A \frac{\partial_l}{\partial \Phi^A} \frac{\partial_l}{\partial \Phi_A^*} X \,, \tag{2.5}$$

where by $A$ in the exponent we mean $A = 0\,(1)$ if $\Phi^A$ is bosonic (fermionic), Einstein summation being understood as operating when there are matching subscript and superscript pairs. On a bosonic functional such as the action itself one can alternatively write

$$\Delta S = \frac{\partial_r}{\partial \Phi^A} \frac{\partial_l}{\partial \Phi_A^*} S \,. \tag{2.6}$$

The measure operator is the quantum part of the QME (as is clear if we restore $\hbar$). Without regularisation it is not well defined. However we will see that the Wilsonian RG provides the regularisation needed in such a way that the relations we review in this and the next subsection are unchanged but then also properly defined.

The QME follows straightforwardly from the observation that

$$\int \mathcal{D}\Phi \, \mathcal{A} \, e^{-S} = \int \mathcal{D}\Phi \, \Delta \, e^{-S} = 0 \,, \tag{2.7}$$

where vanishing follows since the second expression is an integral of a total $\Phi$ derivative.[5] Inherited from its fermionic nature and its Poisson structure, the QME, antibracket and measure operator, satisfy many nice identities, *cf.* [26, 27, 29] and appendix A, which we will use in the following.

## 2.2 BRST cohomology

We will be interested in particular in starting from a solution of the QME (namely the free graviton action) and then perturbing it so that $S + \varepsilon\mathcal{O}$ is still a solution (where $\varepsilon$ is a small parameter, and $\mathcal{O}$ is a quasi-local operator integrated over spacetime). This deforms the BRST algebra, allowing us to explore the space of interacting theories whose gauge invariance is smoothly connected to that of the free one. Substituting the perturbed action into the QME we have that the operator must be BRST invariant:

$$s\,\mathcal{O} = 0 \,, \tag{2.8}$$

---

[4]We write QMF when properties are independent of it being required also to vanish, *i.e.* to satisfy the QME.

[5]In refs. [26, 27] there is an extra step because the antifields are eliminated in the gauge fixing process. We do not need this step because we work in gauge fixed basis and keep the antifields.

where the (full quantum) BRST transformation is:

$$s\,\mathcal{O} = (S, \mathcal{O}) - \Delta \mathcal{O}\,. \tag{2.9}$$

This equation is also valid for fermionic operators (we just choose $\varepsilon$ to be fermionic). We distinguish the *full* BRST transformation from the previously introduced BRST transformation:[6]

$$Q\,\Phi^A = (S, \Phi^A)\,. \tag{2.10}$$

(It is straightforward to see that this is consistent with (2.5) and (2.2). Henneaux *et al* call $Q$ the longitudinal transformation, and $s$ the BRST operator.) We will similarly need the action of the (fermionic) Kozsul–Tate differential [60–62] on the antifields [18–25]:

$$Q^-\Phi_A^* = (S, \Phi_A^*)\,, \tag{2.11}$$

which we thus define also to act from the left. Note that $s\,\Phi^A = Q\,\Phi^A$ and $s\,\Phi_A^* = Q^-\Phi_A^*$ since the quantum part ($\Delta$) automatically vanishes in these cases. Adding these pieces we thus have

$$(Q + Q^-)\,\mathcal{O} = (S, \mathcal{O})\,. \tag{2.12}$$

If $S$ was the classical action $S_{cl}$, which satisfies the $\hbar = 0$ case of the QME, namely the Classical Master Equation:

$$(S_{cl}, S_{cl}) = 0\,, \tag{2.13}$$

then the above would be the full classical BRST transformation, the starting point for classical BRST cohomology [18–25] (see also sec. 2.7). Note however that our charges differ from their classical counterparts because $S \neq S_{cl}$.

The full BRST transformation is nilpotent provided the Master Equation is satisfied. Indeed we have in general (*cf.* [29, 63] and appendix A):

$$s^2\,\mathcal{O} = (\mathcal{A}, \mathcal{O})\,. \tag{2.14}$$

Therefore if the QME is satisfied by $S$, operators that are $s$-exact, *i.e.* can be written as

$$\mathcal{O} = s\,K = (S, K) - \Delta K\,, \tag{2.15}$$

are automatically closed under $s$, *i.e.* satisfy (2.8). However it is evident from the definition of the antibracket and $s$, that such operators just correspond to infinitesimal field and source redefinitions:

$$\delta\Phi^A = \frac{\partial_l K}{\partial \Phi_A^*}\,, \qquad \delta\Phi_A^* = -\frac{\partial_l K}{\partial \Phi^A}\,, \tag{2.16}$$

with $-\Delta K$ corresponding to the Jacobian of the change of variables in the partition function (2.3). Indeed if $\mathcal{O}_1, \cdots, \mathcal{O}_n$ are BRST invariant operators, $\mathcal{O}$ is $s$-exact, and these operators have disjoint spacetime support, then their correlator vanishes:

$$\langle \mathcal{O}\,\mathcal{O}_1 \cdots \mathcal{O}_n \rangle = \langle s\,K\,\mathcal{O}_1 \cdots \mathcal{O}_n \rangle = \langle s\,(K\,\mathcal{O}_1 \cdots \mathcal{O}_n) \rangle = -\frac{1}{\mathcal{Z}} \int \mathcal{D}\Phi\,\Delta\,(K\,\mathcal{O}_1 \cdots \mathcal{O}_n)\,e^{-S} = 0\,. \tag{2.17}$$

Thus if $K$ generates a legitimate change of variables (in particular in our case is quasi-local) then we have to discard this solution as uninteresting. We are therefore interested in operators $\mathcal{O}$ that are closed under $s$ but not exact, *i.e.* in the quantum BRST cohomology.

---

[6]With a loose index, the $\Phi^A$ should be understood in the DeWitt sense as part of an integrated operator.

## 2.3  Wilsonian renormalization group

However operating on local functionals, $\Delta$ in (2.5) is not yet well defined: it needs regularisation. The regularisation that we must use is already dictated by the Wilsonian RG, indeed the bare action without antifields $\mathcal{S} \equiv S^{\text{tot},\Lambda}[\Phi]$ [5], already carries this regularisation. This is the Wilsonian effective action, which is constructed to give the same partition function after integrating out modes with energy scales greater than $\Lambda$. Like the QME, its RG flow is also defined through an integral of a total derivative identity [15, 64, 65]:

$$\partial_t \, \mathrm{e}^{-S} = \frac{\partial_r}{\partial \Phi^A} \left( \hat{\Psi}^A \mathrm{e}^{-S} \right), \tag{2.18}$$

where RG 'time' $t = \ln(\mu/\Lambda)$, $\mu$ some fixed energy scale. For sensible $\hat{\Psi}^A$ and setting $\Phi_A^* = 0$, this defines a valid changed action $S$ which however clearly leaves (2.3) invariant as required. Choosing the field reparametrisation (blocking functional) to be

$$\hat{\Psi}^A = \frac{1}{2} (\dot{\triangle}^\Lambda)^{AB} \frac{\partial_r \Sigma}{\partial \Phi^B}, \tag{2.19}$$

where $\Sigma = S - 2\hat{S}$, $\dot{} \equiv \partial_t$, and $\Delta^\Lambda$ are regularised propagators defined below, gives the RG flow equation:

$$\dot{S} = \frac{1}{2} \frac{\partial_r S}{\partial \Phi^A} (\dot{\triangle}^\Lambda)^{AB} \frac{\partial_l \Sigma}{\partial \Phi^B} - \frac{1}{2} (\dot{\triangle}^\Lambda)^{AB} \frac{\partial_l}{\partial \Phi^B} \frac{\partial_l}{\partial \Phi^A} \Sigma. \tag{2.20}$$

If the seed action $\hat{S}$ is chosen to coincide with the Gaussian fixed point action:

$$\mathcal{S}_0 = \frac{1}{2} \Phi^A (\triangle^\Lambda)_{AB}^{-1} \Phi^B, \tag{2.21}$$

then (2.20) is the Wilson/Polchinski equation [66, 67]. It is convenient to summarise (2.20) in the notation [68]:

$$\dot{S} = a_0[S, \Sigma] - a_1[\Sigma], \tag{2.22}$$

where the classical piece, $a_0$, is thus symmetric bilinear in its arguments, and the quantum piece, $a_1$, is linear in its arguments. First order perturbations of $S$ are thus operators $\mathcal{O}$ (integrated over spacetime) whose RG flow is governed by

$$\dot{\mathcal{O}} = 2a_0[\mathcal{O}, S - \hat{S}] - a_1[\mathcal{O}]. \tag{2.23}$$

In particular if $S$ is a fixed point action, solving this equation by separation of variables, gives the eigenoperators.

The Gaussian fixed point corresponds to a regulated massless free field theory. Its propagators (which exist since we are working in the gauge fixed basis),

$$\triangle^{AB} = (-)^A \triangle^{BA} = (-)^B \triangle^{BA}, \tag{2.24}$$

are regularised by multiplying by an ultraviolet cutoff function $C^\Lambda(p) \equiv C(p^2/\Lambda^2)$, so that[7]

$$(\triangle^\Lambda)^{AB} = C^\Lambda(p) \triangle^{AB}, \tag{2.25}$$

Thus in (2.21), the inverse propagators $\Delta_{AB}^{-1}$ carry a factor of $1/C^\Lambda(p)$. Qualitatively, for $|p| < \Lambda$, $C^\Lambda(p) \approx 1$ and mostly leaves the modes unaffected, while for $|p| > \Lambda$ its rôle is to suppress modes. We require that $C(p^2/\Lambda^2)$ is a smooth monotonically decreasing function of its argument, that $C^\Lambda(p) \to 1$ for $|p|/\Lambda \to 0$, and for $|p|/\Lambda \to \infty$ we require that $C^\Lambda(p) \to 0$ sufficiently fast to ensure that all momentum integrals are regulated in the ultraviolet. $S = \mathcal{S}_0$ is a solution of the RG flow equation (2.20) (after throwing away a field independent term), i.e. satisfies $\dot{\mathcal{S}}_0 = -a_0[\mathcal{S}_0, \mathcal{S}_0]$. If we had scaled to dimensionless variables by using $\Lambda$, it would indeed be unchanged (a fixed point) as further modes are integrated out.

---

[7]The propagators are diagonal in the momentum $p$. Here and similarly later, we trust the reader understands our slight abuse of notation.

### 2.4 Combining the concepts

In order to combine these concepts, we need them to match in particular at the Gaussian fixed point and for first order perturbations away from this fixed point. We will see that this fixes uniquely how antifield dependence is to be introduced into the Wilsonian RG. We also need the QME to be respected by the RG flow, *i.e.* we require

$$\mathcal{A}[S] = 0 \quad \Longrightarrow \quad \partial_t \mathcal{A}[S] = 0, \tag{2.26}$$

so that if the QME is satisfied at some scale $\Lambda$, it remains satisfied on further RG evolution. We will see that this fixes uniquely how the parts of the QMF, (2.5), are to be regularised.

Since the Gaussian fixed point corresponds to the free action, we have there only the free BRST transformations generated by $Q_0$, where by

$$Q_0 \Phi^A = R^A_{\ B} \Phi^B, \tag{2.27}$$

we mean furthermore the classical (unregularised) transformations. Since the free BRST symmetry is Abelian and diagonal in momentum space, it is not difficult to regularise it consistently by inserting some momentum cutoff function dependence between the bilinear terms in the action and between the functional derivatives in (2.5). Thus from (2.2) we now write

$$S_0 = \mathcal{S}_0 + \mathcal{S}_0^*, \tag{2.28}$$

where

$$\mathcal{S}_0^* = -(Q_0 \Phi^A) B^\Lambda \Phi_A^*, \tag{2.29}$$

and $B^\Lambda(p)$ is some cutoff function dependence to be determined. For this to make sense under the RG, $S_0$ must also be a solution of the flow equation. By converting to dimensionless variables it will then be a fixed point action, now with antifields included. Plugging $S = S_0$ into the flow equation (2.22) gives:

$$\dot{\mathcal{S}}_0 + \dot{\mathcal{S}}_0^* = -a_0[\mathcal{S}_0, \mathcal{S}_0] + a_0[\mathcal{S}_0^*, \mathcal{S}_0^*]. \tag{2.30}$$

At first sight the structure (2.28) cannot be preserved since *a priori* the last term above generates terms bilinear in the antifields. In fact up to a multiplicative factor (containing the cutoff functions) $a_0[\mathcal{S}_0^*, \mathcal{S}_0^*]$ computes

$$\left\langle Q_0 \Phi^A Q_0 \Phi^B \right\rangle = \left\langle Q_0 \left( \Phi^A Q_0 \Phi^B \right) \right\rangle = 0, \tag{2.31}$$

which vanishes by BRST invariance. Thus the flow equation reduces to $\dot{\mathcal{S}}_0^* = 0$, which would tell us without loss of generality to set $B^\Lambda \equiv 1$.

However first order perturbations will lead us to a better solution. Expanding in the gauge coupling, let us call it $\kappa$, as:

$$S = S_0 + \kappa S_1 + \tfrac{1}{2} \kappa^2 S_2 + \cdots, \tag{2.32}$$

we have from (2.23) that first order perturbations satisfy

$$\dot{S}_1 = 2a_0[S_1, S_0 - \hat{S}] - a_1[S_1]. \tag{2.33}$$

We see that even the original $S_1 = \mathcal{O}[\Phi]$ eigenoperators, *i.e.* for gravity those given in (1.4), must now inherit antifield dependence as dictated by the first term on the right hand side, unless we insist (as we now do) that the seed action is also modified so as to maintain equality with the Gaussian fixed point action (2.28), *i.e.*

$$\hat{S} = \mathcal{S}_0 + \mathcal{S}_0^*. \tag{2.34}$$

Checking again that $S_0$ satisfies the flow equation we now have

$$\dot{\mathcal{S}}_0 + \dot{\mathcal{S}}_0^* = -a_0[\mathcal{S}_0, \mathcal{S}_0] - 2a_0[\mathcal{S}_0, \mathcal{S}_0^*], \tag{2.35}$$

and thus find that actually in (2.29) we should set $B^\Lambda(p) = 1/C^\Lambda(p)$. We conclude that the free action (2.28) and seed action (2.34) are equal, and regularised by inserting $1/C^\Lambda(p)$ inside all bilinear terms.

To find out how the QME is to be regularised, we insist on (2.26). Inserting some cutoff function dependence (to be determined) between the functional derivatives in $\Delta$, and writing $\mu = e^{-S}$, we have

$$\begin{aligned}
\partial_t(\mathcal{A}\mu) &= \dot{\Delta}\mu + \Delta\dot{\mu} \\
&= \dot{\Delta}\mu + \frac{1}{2}\frac{\partial_r}{\partial\Phi^A}\left\{(\dot{\triangle}^\Lambda)^{AB}\frac{\partial_r\Delta S}{\partial\Phi^B}\mu + (\dot{\triangle}^\Lambda)^{BA}\frac{\partial_r\Sigma}{\partial\Phi^B}\Delta\mu + (\dot{\triangle}^\Lambda)^{BA}\left(\frac{\partial_r\Sigma}{\partial\Phi^B},\mu\right)\right\} \\
&= \dot{\Delta}\mu - \frac{\partial_r}{\partial\Phi^A}(\dot{\triangle}^\Lambda)^{BA}\left(\frac{\partial_r\hat{S}}{\partial\Phi^B},\mu\right) + \frac{1}{2}\frac{\partial_r}{\partial\Phi^A}\left\{(\dot{\triangle}^\Lambda)^{BA}\frac{\partial_r\Sigma}{\partial\Phi^B}\mathcal{A}\mu - (\dot{\triangle}^\Lambda)^{AB}\frac{\partial_r\mathcal{A}}{\partial\Phi^B}\mu\right\},
\end{aligned} \tag{2.36}$$

where in the first line we use the first equality in (2.7) and recognise that $\Delta$ now has RG time dependence, in the second line we insert (2.20), use (A.4), and for prettiness (2.24), and recognise that $\hat{S}$ is only bilinear. In the final line we use (A.6).

We see that the consistency relation (2.26) will be satisfied only if we can get rid of the first two terms in the last line. Substituting (2.34) in the second term and expanding, the $\mathcal{S}_0^*$ part gives

$$-\frac{1}{2}\left\{R^C_{\phantom{C}B}(\dot{\triangle}^\Lambda)^{BA} + R^A_{\phantom{A}B}(\dot{\triangle}^\Lambda)^{BC}\right\}\frac{\partial_l^2\mu}{\partial\Phi^A\partial\Phi^C}. \tag{2.37}$$

The term in braces vanishes by linearised BRST invariance, as shown in (A.8). That leaves the $\mathcal{S}_0$ part, which one quickly finds cancels the first term if and only if $\Delta$ is regularised by inserting $C^\Lambda(p)$ between its functional derivatives. We set this to be true from now on.

Pulling out $\mu$ as an overall factor, and cancelling terms using (2.20), (2.36) then becomes

$$\dot{\mathcal{A}} = 2a_0[\mathcal{A}, S - \hat{S}] - a_1[\mathcal{A}], \tag{2.38}$$

*i.e.* the QME, $\mathcal{A}$, simply satisfies the operator flow equation (2.23), and thus clearly also (2.26).

The framework we have derived has nice properties, as we explain in sec. 2.6.

## 2.5 Comparison with earlier work

Before doing so, it is helpful to compare our framework to earlier work. Had we stuck with the solution $B^\Lambda = 1$ in (2.29), as we found below (2.31), we would still require the QME to be consistent with the RG flow. This gives us the same equation as (2.36). Although in this $B^\Lambda = 1$ solution, $\hat{S}$ has no $\Phi^*$ dependence, we just saw that consistency is independent of that part, since (2.37) vanishes. Thus we learn again that the QME is to be regularised by inserting $C^\Lambda$.

This $B^\Lambda = 1$ formulation coincides with those of refs. [40–42, 47, 48], and as we will see shortly is actually related to our formulation by a change of variables. It has however the unpleasant feature that from (2.10) already at the free level, the regularised BRST transformations

$$Q_0\Phi^A|_{\text{reg}} = (S_0, \Phi^A) = C^\Lambda(p)R^A_{\phantom{A}B}\Phi^B \tag{2.39}$$

are no longer local but are now only quasi-local (*cf.* footnote 2) and similarly from (2.11) also the free Koszul–Tate differential is now only quasi-local:

$$Q_0^-\Phi_A^*|_{\text{reg}} = (S_0, \Phi_A^*) = C^\Lambda(p)Q_0^-\Phi_A^*. \tag{2.40}$$

This causes difficulties particularly in understanding the quantum BRST cohomology, since this means that it cannot close on local terms. As we note in sec. 2.6 this problem is absent from our formulation.

Motivated by the wish to preserve the canonical structure

$$(\Phi^A, \Phi_B^*) = \delta_B^A, \tag{2.41}$$

which will then also remove the above issue, the authors in refs. [43,47] also changed variables

$$\Phi^*(p) \mapsto C^\Lambda(p)\Phi^*(p). \tag{2.42}$$

Even though this is just a change of variables, it obscures RG properties since the partition function (2.3) now depends on $\Lambda$. Thus the flow equation no longer takes the form (2.18). Most importantly, it obscures how $\Delta$ is to be regularised since it removes the cutoff dependence from here also. Thus in this parametrisation it is no longer true that $\Delta$ is well defined when acting on local functionals.

From sec. 2.4, we already know that another feature of both of the above formulations is that they lead to extra antifield dependence, even for the original $S_1 = \mathcal{O}[\Phi]$ eigenoperators. Working with their first formulation, *i.e.* without (2.42), and using the fact that $S_0 - \hat{S} = \mathcal{S}_0^*$ with $B^\Lambda = 1$, (2.33) tells us (*e.g.* by adapting the method of characteristics) that the general solution is given by an $S_1 \equiv S_1[\check{\Phi}, \Phi^*]$ which satisfies the eigenoperator equation without the antifield correction (just as it does in our formulation):

$$\dot{S}_1[\check{\Phi}, \Phi^*] = -a_1[S_1], \tag{2.43}$$

its effect being carried by:

$$\check{\Phi}^A = \Phi^A + \triangle_\Lambda^{AB} R_B^C \Phi_C^*. \tag{2.44}$$

The requirement of quasi-locality fixes the $t$-integration constant to give the infrared regulated propagator [5]

$$\triangle_\Lambda^{AB} = C_\Lambda(p)\triangle^{AB} \qquad \text{where} \qquad C_\Lambda(p) = 1 - C^\Lambda(p). \tag{2.45}$$

These $\check{\Phi}^A$ coincide with the shifted fields found in [44, 45, 47].

However if we take the hint from the mathematics, and fully transform the equations to shifted variables, we get back to our formulation. Firstly, we can show that (2.44) is a finite quantum canonical transformation, *i.e.* leaves invariant the QME. This follows because it can be written as:

$$\check{\Phi}^A = \frac{\partial_l}{\partial \check{\Phi}_A^*} K[\Phi, \check{\Phi}^*], \qquad \Phi_A^* = \frac{\partial_r}{\partial \Phi^A} K[\Phi, \check{\Phi}^*], \tag{2.46}$$

where

$$K = \check{\Phi}_A^* \Phi^A + \Psi^*[\check{\Phi}^*], \qquad \text{and} \qquad \Psi^*[\check{\Phi}^*] = \tfrac{1}{2} \check{\Phi}_A^* \triangle_\Lambda^{AB} R_B^C \check{\Phi}_C^*. \tag{2.47}$$

*cf.* appendix A for more details and *e.g.* ref. [29]. Secondly, substituting the transformation (2.44) into (2.28), and noting that the $(\Phi^*)^2$ terms vanish for the same reasons as in (2.31), one finds that $S_0[\check{\Phi}, \Phi^*]$ takes again the same form, but with $B^\Lambda = 1$ replaced by $B^\Lambda = 1/C^\Lambda$ in (2.29). Similarly the seed action, which was just (2.21), becomes $\hat{S} = S_0 - (Q_0 \check{\Phi}^A)\Phi_A^*$. Substituting the transformation (2.44) into the flow equation (2.22), we see that this second piece of $\hat{S}$ is cancelled by the change from $\partial_t|_\Phi$ to $\partial_t|_{\check{\Phi}}$, just as happened for the operators in the passage from (2.33) to (2.43). Thus the net result is we get back our flow equation *i.e.* with the seed action and Gaussian action now set equal and regulated by inserting $1/C^\Lambda$ in all terms.

### 2.6 Summary and further properties

To summarise our formulation, the free action (2.28) (equal to the seed action) is regularised by inserting $1/C^\Lambda(p)$ inside all bilinear terms:

$$S_0 = \hat{S} = \mathcal{S}_0 + \mathcal{S}_0^* = \tfrac{1}{2}\,\Phi^A(\triangle^\Lambda)^{-1}_{AB}\Phi^B - (Q_0\Phi^A)\left(C^\Lambda\right)^{-1}\Phi_A^* . \tag{2.48}$$

The QMF (2.4) is regularised by inserting $C^\Lambda(p)$ between the functional derivatives in (2.5):

$$(X,Y) = \frac{\partial_r X}{\partial \Phi^A}\,C^\Lambda\,\frac{\partial_l Y}{\partial \Phi_A^*} - \frac{\partial_r X}{\partial \Phi_A^*}\,C^\Lambda\,\frac{\partial_l Y}{\partial \Phi^A} \qquad \text{and} \qquad \Delta X = (-)^A\frac{\partial_l}{\partial \Phi^A}\,C^\Lambda\,\frac{\partial_l}{\partial \Phi_A^*}X , \tag{2.49}$$

Note that this means the canonical structure (2.41) is not preserved. Since the same regularisation must be applied to the antibracket and the measure term, so that the identity (2.7) remains satisfied, some sort of deformation of (2.41) is a necessary consequence if $\Delta$ is to be well defined when acting on arbitrary local functionals. This is related to the observation [26] that $\Delta$ quantifies the extent to which the volume of phase space is not preserved by a canonical transformation.

This is not a problem however because what matters are the BRST transformations themselves. Using the definition (2.10), factors of cutoff and its inverse actually cancel each other in the free BRST transformation

$$Q_0\Phi^A = \left(S_0, \Phi^A\right) , \tag{2.50}$$

so that it is left unaltered by the regularisation. Clearly this is also true of the free Kozsul–Tate differential where, by (2.11),

$$Q_0^-\Phi_A^* = \left(S_0, \Phi_A^*\right) . \tag{2.51}$$

Thus only the quantum part of the free BRST cohomology, $\Delta$, depends on the cutoff, and in a way which is well defined (*i.e.* regularised) when acting on arbitrary local functionals, provided we use a suitably fast decaying $C^\Lambda$ *e.g.* exponential as we already used in refs. [5,8]. Since all the relations that we need from the QME follow from (2.7) and from symmetry and statistics, in particular the BRST cohomology relations in sec. 2.2, it straightforward to see that all these are left undisturbed by the regularisation. Since $\Delta$ also maps local functionals to local functionals, this means that in this formulation the free quantum BRST cohomology is now well defined on local functionals.

Substituting the perturbative expansion of $S$, (2.32), into the full BRST differential (2.9) gives its perturbative expansion $s = s_0 + \kappa s_1 + \tfrac{1}{2}\kappa^2 s_2 + \cdots$. Non-trivial solutions of the free quantum BRST cohomology define all possible perturbative interactions to first order through the relation:

$$s_0 S_1 = 0 . \tag{2.52}$$

We have just seen that this can be studied in the space of local functionals. Provided that there are no obstructions, the higher orders can be iteratively constructed by substituting (2.32) into the QME (2.4) (the measure operator appears only through $s_0$ on the left hand side):

$$s_0 S_2 = -\tfrac{1}{2}(S_1, S_1), \qquad s_0 S_3 = -(S_1, S_2), \qquad \cdots . \tag{2.53}$$

The ambiguities in the solution of the higher order pieces are therefore again elements of the free BRST cohomology. However note that if $S_1$ is local, the particular solutions for the $S_{n>1}$ will be only quasi-local, because of the presence of $C^\Lambda$ in the definition of the antibracket on the right hand sides.

Note that it is the free action together with its antifield dependence, *i.e.* (2.28), that is the Gaussian fixed point. The part with no antifields, (2.21), is no longer separately a solution of

the flow equation. Since $\hat{S} = S_0$ is maintained after introducing antifields, general first order perturbations $S_1$ (with or without antifields) continue to satisfy the simple equation:

$$\dot{S}_1 = -a_1[S_1]. \tag{2.54}$$

This just tells us that $S_1$ must be a linear combination of eigen-operators with constant coefficients (the couplings). As we show in ensuing sections, if we insist that the eigen-operators span a space of interactions closed under the Wilsonian RG, this in turn means that $S_1$ is an element of the Hilbert space $\mathfrak{L}$, defined by (1.3) and (1.4) but extended to include the ghost, auxiliary, and anti-ghost fields.

The full action $S$ satisfies the flow equation (2.20), or in short-hand (2.22). Suppose that we shift $S$ infinitesimally to a new solution $S + \epsilon K$. It is straightforward to confirm that also if $\epsilon$ is fermionic, $K$ satisfies the operator flow equation (2.23). On the other hand the QMF also satisfies the operator flow equation, as we saw in (2.38). Under the infinitesimal shift, the QMF becomes $\mathcal{A} - \epsilon\, sK$, as follows from (2.4) and (2.9), and therefore also $sK$ satisfies (2.23). We have thus shown that if $K$ satisfies the operator flow equation, so does $sK$. In particular around the Gaussian fixed point $S = S_0$, the operator flow equation is just (2.54). Therefore we have shown that if $K$ is a linear combination of eigenoperators with constant coefficients (the couplings), then the cohomologically trivial operator $\mathcal{O} = s_0 K$, cf. (2.15), is also a linear combination of eigenoperators with constant coefficients. Since, as we will see, the eigen-operators are also local functionals, we see that the free quantum BRST cohomology can be defined within the Hilbert space $\mathfrak{L}$ spanned by these eigen-operators with their constant couplings.

Clearly this is precisely what we need to understand when searching for non-trivial continuum limits that incorporate diffeomorphism invariance. We have thus arrived at the ideal framework for developing the quantum BRST cohomology, and as close to ideal as possible more generally for quantum calculations where power law divergences need to be kept under rigorous control.

Finally we note that since we still have $\hat{S} = S_0$, the general flow equation for the action and operators is more simply expressed in terms of the interactions only. Substituting $S = S_0 + S^{\text{int}}$ into the flow equation (2.22), and discarding the field independent piece, turns (2.20) into recognisably the usual form for the Polchinski flow equation [66]:

$$\dot{S}^{\text{int}} = a_0[S^{\text{int}}, S^{\text{int}}] - a_1[S^{\text{int}}] = \frac{1}{2}\frac{\partial_r S^{\text{int}}}{\partial \Phi^A}(\dot{\triangle}^\wedge)^{AB}\frac{\partial_l S^{\text{int}}}{\partial \Phi^B} - \frac{1}{2}(\dot{\triangle}^\wedge)^{AB}\frac{\partial_l}{\partial \Phi^B}\frac{\partial_l}{\partial \Phi^A}S^{\text{int}}, \tag{2.55}$$

while from (2.23), we have

$$\dot{\mathcal{O}} = 2a_0[\mathcal{O}, S^{\text{int}}] - a_1[\mathcal{O}]. \tag{2.56}$$

## 2.7 Gauge invariant basis

Up until now we have tacitly been working in the gauge fixed basis [29, 55–59] where propagators are well defined, while also keeping the anti-fields as sources for the BRST variations. This is the starting point for analysis of renormalizability when gauge invariance is involved [54].

Expressions are simpler in the gauge invariant basis however, i.e. the system before any gauge fixing is applied. Anti-fields are still included to encode the action of the BRST complex, i.e. including also the Kozsul-Tate operator and at the quantum level the measure operator $\Delta$. At the classical level and working over the space of strictly local operators, this is the starting point for the study of classical BRST cohomology [18–22, 24, 25].

The two bases are in fact related by a quantum canonical transformation. If we let the gauge invariant basis fields be $\{\check{\Phi}^A, \check{\Phi}^*_A\}$, then the quantum canonical transformation again takes the same form (2.46), however in this case

$$K = \check{\Phi}^*_A \Phi^A + \Psi[\Phi], \tag{2.57}$$

where $\Psi$ is the so-called gauge fixing fermion [26, 27]. Since $\Psi[\Phi]$ depends only on the fields and not the antifields, a straightforward adaptation of the arguments in appendix A establishes that $K$ indeed generates a quantum canonical transformation.

This leads to a slightly subtle point. It means that results, even at the quantum level, are equivalent when calculated in either basis. This does not mean however that it furnishes a way to compute general quantum corrections without gauge fixing. In this framework, the general rules for calculating quantum corrections have to be derived in a given gauge fixed basis, so that propagators exist. By the canonical transformation, this implies a set of rules for computation in the gauge invariant basis, which will however still be tied to the chosen gauge fixing; for example the propagators are still the ones derived in the chosen gauge fixed basis. Although the QMF is invariant under the transformation to the gauge invariant basis, the flow equation (2.20) is not. Since its form in the gauge invariant basis is unilluminating, we do not display it.

On the other hand the form of the quantum-corrected BRST transformations follow from the QME. For this we need only the regularised measure operator $\Delta$. We do not need the propagators. Thus to study the quantum BRST cohomology, we are free to use the gauge invariant basis. Evidently this leads to simplifications. In fact it will allow us to work in the minimal basis [26, 27]. This also has the advantage that the properties we derive will clearly continue to hold whatever legitimate gauge fixing we then implement for computing the actual quantum corrections from the Wilsonian RG.

# 3 Quantum Gravity

We now specialise the general structure we have derived in the previous section to that of quantum gravity, and take the opportunity to review and extend the analysis in ref. [5], in particular to the ghost and auxiliary fields. We also generalise it to $d$ dimensions, in case this will prove useful in future, and begin by working in a more general gauge. Our purpose in these sections is not only to prepare for the quantum BRST cohomology study in sec. 7.2, but also to prepare the ground for future calculations in this renormalizable quantum gravity theory.

We start with the BRST algebra in gauge invariant basis. As we discuss at the end of sec. 3.2, by comparison with the algebra in (a convenient) gauge fixed basis, it is significantly simpler. Since the two bases are equivalent under a canonical transformation, it makes sense to study the BRST algebra exclusively in the gauge invariant basis. Then as we already pointed out in sec. 2.7, also the BRST cohomology is clearly independent of the gauge choice. We note that for the BRST cohomology, we can also choose to work within the minimal basis, which then leads to yet further simplifications.

## 3.1 Quantum gravity BRST algebra in gauge invariant basis

To make sense of the Wilsonian RG for quantum gravity, we need to work in Euclidean signature and around flat $\mathbb{R}^d$ [5]. Then the action for free graviton fields $H_{\mu\nu}$ is:

$$\mathcal{S}_0 = \int d^d x \, \mathcal{L}_0 \, , \quad \text{where} \quad \mathcal{L}_0[H] = \frac{1}{2} \left( \partial_\lambda H_{\mu\nu} \right)^2 - 2 \left( \partial_\lambda \varphi \right)^2 - \left( \partial^\mu H_{\mu\nu} \right)^2 + 2 \, \partial^\alpha \varphi \, \partial^\beta H_{\alpha\beta} \, . \quad (3.1)$$

We have written the trace as $\varphi = \frac{1}{2} H_{\mu\mu}$, and contraction is with the flat metric $\delta_{\mu\nu}$. Since raising an index thus makes no difference we will usually leave all indices as subscripts. Given that the action is normalised, bilinear, and quadratic in derivatives, it is determined uniquely

by linearised diffeomorphism invariance, or in BRST language:

$$Q_0 H_{\mu\nu} = \partial_\mu c_\nu + \partial_\nu c_\mu, \tag{3.2}$$

where $c_\mu$ are the ghost fields.

The free system, *i.e.* the action (3.1) and invariance (3.2), follow from the Einstein-Hilbert Lagrangian

$$\mathcal{L}_{EH} = -2\sqrt{g}R/\kappa^2, \tag{3.3}$$

if one writes the metric to $O(\kappa)$, as

$$g_{\mu\nu} = \delta_{\mu\nu} + \kappa H_{\mu\nu} \tag{3.4}$$

(and to get $\kappa$ powers correct, regard $\kappa c^\mu$ as the small diffeomorphism). However one of the main points of the paper, and in particular sec. 7.2, is to determine the constraints on alternative quantum deformations of the free system, thus extending to the quantum domain the questions asked of classical BRST cohomology. Therefore consistent interactions need not *a priori* correspond to those that arise from $\mathcal{L}_{EH}$.

Recall from the previous section that the free action is regulated by inserting $1/C^\Lambda(p)$ between the two fields in all terms, *cf.* (2.48). Therefore we now write

$$\mathcal{L}_0 = \tfrac{1}{2} H_{\mu\nu} (\triangle^\Lambda)^{-1}_{\mu\nu,\alpha\beta} H_{\alpha\beta}, \tag{3.5}$$

where $\triangle^{-1}_{H_{\mu\nu}H_{\alpha\beta}} = \triangle^{-1}_{\mu\nu,\alpha\beta}$ is the differential operator we get from (3.1) by integrating by parts, and $(\triangle^\Lambda)^{-1}_{\mu\nu,\alpha\beta} = \triangle^{-1}_{\mu\nu,\alpha\beta}/C^\Lambda$. To encode the BRST complex and its deformations, we add to the action the antifield source terms. At this stage we only need the fermionic symmetric tensor $H^*_{\mu\nu}(x)$:

$$S_0 = \int d^d x\, L_0, \quad \text{where} \quad L_0 = \mathcal{L}_0 - 2\,\partial_\mu c_\nu \left(C^\Lambda\right)^{-1} H^*_{\mu\nu}, \tag{3.6}$$

where the structure follows (2.2). This is the action at $O(\kappa^0)$ in the *minimal gauge invariant basis*. This basis encodes all the properties of the gauge invariant action and the gauge transformations. The antighost $c^*_\mu$ is conjugate to the commutator of gauge transformations. Since at the free level, these transformations (3.2) are Abelian, $c^*_\mu$ does not appear. This will change when we introduce interactions, as for example in the standard realisation of diffeomorphism invariance reviewed in sec. 7.1.

To get the *non-minimal gauge invariant basis* we introduce the bosonic auxiliary field $b_\mu$ and the bosonic anti-ghost anti-field $\bar{c}^*_\mu$ and write:

$$L_0 = \mathcal{L}_0 + \frac{1}{2\alpha}\, b_\mu \left(C^\Lambda\right)^{-1} b_\mu - 2\,\partial_\mu c_\nu \left(C^\Lambda\right)^{-1} H^*_{\mu\nu} - i\, b_\mu \left(C^\Lambda\right)^{-1} \bar{c}^*_\mu, \tag{3.7}$$

where $\alpha$ will become our gauge fixing parameter. As we will see in the next section, only after mapping to gauge fixed basis will we get dependence on $\bar{c}_\mu$ at the free level.

Recall that we also insert $C^\Lambda(p)$ between the pairs in the antibracket, as in (2.49). Thus using the general formula (2.50), we verify (3.2), and confirm that it is the only non-vanishing free BRST transformation in minimal basis (*i.e.* $Q_0 c_\mu = 0$). We also read off the one further non-vanishing free BRST transformation that appears in the non-minimal basis:

$$Q_0 \bar{c}_\mu = i\, b_\mu. \tag{3.8}$$

Similarly from (2.51), we read off the non-vanishing free Kozsul-Tate differentials (both of which are already present in the minimal basis):

$$Q^-_0 H^*_{\mu\nu} = -2 G^{(1)}_{\mu\nu}, \qquad Q^-_0 c^*_\nu = -2 \partial_\mu H^*_{\mu\nu}, \tag{3.9}$$

where $G^{(1)}_{\mu\nu}$ is the linearised Einstein tensor:

$$G^{(1)}_{\mu\nu} = -R^{(1)}_{\mu\nu} + \tfrac{1}{2}R^{(1)}\delta_{\mu\nu} = \tfrac{1}{2}\Box H_{\mu\nu} - \delta_{\mu\nu}\Box\varphi + \partial^2_{\mu\nu}\varphi + \tfrac{1}{2}\delta_{\mu\nu}\partial^2_{\alpha\beta}H_{\alpha\beta} - \partial_{(\mu}\partial^\alpha H_{\nu)\alpha}, \quad (3.10)$$

the linearised curvatures being[8]

$$R^{(1)}_{\mu\alpha\nu\beta} = -2\partial_{[\mu|}\partial_{[\nu}H_{\beta]|\alpha]}, \ R^{(1)}_{\mu\nu} = -\partial^2_{\mu\nu}\varphi + \partial_{(\mu}\partial^\alpha H_{\nu)\alpha} - \tfrac{1}{2}\Box H_{\mu\nu}, \ R^{(1)} = \partial^2_{\alpha\beta}H_{\alpha\beta} - 2\Box\varphi.$$
$$(3.11)$$

As we noted in sec. 2.6, the above transformations are unaltered by the regularisation, in particular $G^{(1)}_{\mu\nu}$ is free of regularisation.

As well as ghost number, and statistics, the system carries another natural grading [18,23] which can be thought of, depending on the field, as the antifield number or the antighost number. Assigning $S$ zero ghost number, $c_\mu$ unit ghost number, and $H^*_{\mu\nu}$ unit antifield number, consistent assignments for all other fields and operators demands the values given in table 1, where we also display the Grassmann grading and their engineering dimension. (We do not list the antifield $b^*_\mu$ because this never appears in the action.)

Table 1: The various Abelian charges (a.k.a. gradings) carried by the fields and operators. $\epsilon$ is the Grassmann grading, being 1(0) if the object is fermionic (bosonic). gh # is the ghost number, ag # the antighost/antifield number, pure gh # = gh # + ag #, and dimension is the engineering dimension. The first two rows are the minimal set of fields, the next two make it up to the non-minimal set, then the ensuing two rows are the minimal set of antifields, and $\bar{c}^*_\mu$ is needed for the non-minimal set. Finally, the charges are determined in order to ensure that $Q$ and $Q^-$ can also be assigned definite charges.

|  | $\epsilon$ | gh # | ag # | pure gh # | dimension |
|---|---|---|---|---|---|
| $H_{\mu\nu}$ | 0 | 0 | 0 | 0 | $(d-2)/2$ |
| $c_\mu$ | 1 | 1 | 0 | 1 | $(d-2)/2$ |
| $\bar{c}_\mu$ | 1 | -1 | 1 | 0 | $(d-2)/2$ |
| $b_\mu$ | 0 | 0 | 1 | 1 | $d/2$ |
| $H^*_{\mu\nu}$ | 1 | -1 | 1 | 0 | $d/2$ |
| $c^*_\mu$ | 0 | -2 | 2 | 0 | $d/2$ |
| $\bar{c}^*_\mu$ | 0 | 0 | 0 | 0 | $d/2$ |
| $Q$ | 1 | 1 | 0 | 1 | 1 |
| $Q^-$ | 1 | 1 | -1 | 0 | 0 |

According to the assignments in table 1, one also sees that $(X, Y)$ adds one to the sum of the dimensions of $X$ and $Y$, and adds one to the sum of the ghost numbers of $X$ and $Y$. Therefore both charges $Q$ and $Q^-$ increase the ghost number and dimension by one. Similarly $\Delta$ increases ghost number and dimension by one. Finally we note that a canonical transformation $K$ as in (2.46), must thus be fermionic and have ghost number $-1$.

Although the action $S$ has definite ghost number, namely vanishing ghost number, it does not have definite anti-field number. Following refs. [18–25], we can therefore split a BRST cohomology problem into parts depending on the anti-field number, labelling the parts with a superscript indicating the anti-field/anti-ghost number: $S = \sum_{n=0} S^n$.

---

[8]defining symmetrisation as: $t_{(\mu\nu)} = \tfrac{1}{2}(t_{\mu\nu} + t_{\nu\mu})$, and antisymmetrisation as $t_{[\mu\nu]} = \tfrac{1}{2}(t_{\mu\nu} - t_{\nu\mu})$.

Since the BRST charge leaves the antighost number undisturbed we write it as $Q \equiv Q^0$. On the other hand the Kozsul-Tate charge decreases antighost number by one, which is why we label it as $Q^-$, *i.e.* with a minus in the superscript. The measure operator $\Delta = \Delta^- + \Delta^=$ can also be divided into parts of definite antighost number:

$$\Delta^- = \frac{\partial}{\partial H_{\mu\nu}} C^\Lambda \frac{\partial_l}{\partial H^*_{\mu\nu}} - \frac{\partial_l}{\partial \bar{c}_\mu} C^\Lambda \frac{\partial}{\partial \bar{c}^*_\mu}, \qquad \Delta^= = -\frac{\partial_l}{\partial c_\mu} C^\Lambda \frac{\partial}{\partial c^*_\mu}. \tag{3.12}$$

Thus $\Delta^-$ lowers the antighost number by one, and $\Delta^=$ lowers it by two. (On the minimal set, only the first piece of $\Delta^-$ is active.) The full quantum BRST charge can now be written as

$$s = Q + Q^- - \Delta^- - \Delta^=. \tag{3.13}$$

Splitting the cohomology problem $s\mathcal{O} = 0$ by antighost number, this becomes the statement that for $n \geq 0$ the following equations must be satisfied:

$$Q\,\mathcal{O}^n + (Q^- - \Delta^-)\mathcal{O}^{n+1} - \Delta^= \mathcal{O}^{n+2} = 0. \tag{3.14}$$

Since we require the QME, *cf.* sec. 2.2, we want the solutions modulo the trivial (exact) ones, $\mathcal{O} = sK$. Grading $K$ also by antighost number, these trivial solutions take the form:

$$\mathcal{O}^n = Q\,K^n + (Q^- - \Delta^-)K^{n+1} - \Delta^= K^{n+2}, \tag{3.15}$$

such that the $K^n$ are fermionic and have ghost number $-1$. Grading $s^2$ by antighost number it is almost immediate to see that:

$$Q^2 = 0, \quad (Q^-)^2 = 0, \quad (\Delta^-)^2 = 0, \quad (\Delta^=)^2 = 0,$$
$$\{Q, Q^-\} = 0, \quad \{Q, \Delta^-\} = 0, \quad \{Q^-, \Delta^=\} = 0, \quad \{\Delta^-, \Delta^=\} = 0,$$
$$\{Q^-, \Delta^-\} + \{Q, \Delta^=\} = 0. \tag{3.16}$$

Thus $Q^2$ is the only piece that leaves antighost number unchanged and therefore the quantum BRST charge $Q$ must be nilpotent on its own. The anticommutators involving only $\Delta$ must vanish because functional derivatives (anti)commute and $\Delta$ is overall odd. Then the anticommutator $\{Q^-, \Delta^=\}$ must vanish since it is the only remaining operator that lowers antighost number by three. The piece that lowers antighost number by one must vanish on its own:

$$\{Q, Q^-\} - \{Q, \Delta^-\} = 0. \tag{3.17}$$

Again since the operators are odd, each anticommutator can only be non-vanishing if a functional derivative is used up by acting on the other operator. Then the first anticommutator contains precisely one free functional derivative, and the second contains precisely two. As an operator identity, they must therefore vanish separately. The remaining pieces lower antighost number by two:

$$(Q^-)^2 - \{Q^-, \Delta^-\} - \{Q, \Delta^=\} = 0. \tag{3.18}$$

By the same argument we see that $(Q^-)^2 = 0$. The final two pieces do not vanish separately. Indeed by explicit computation even at the free level the result is non-vanishing:

$$\{Q_0, \Delta^=\} = -\{Q_0^-, \Delta^-\} = 2\frac{\partial}{\partial c^*_\nu} C^\Lambda \partial_\mu \frac{\partial}{\partial H_{\mu\nu}}. \tag{3.19}$$

### 3.2 Quantum gravity BRST algebra in a gauge fixed basis

Gauge fixing is implemented by a suitable gauge fixing fermion $\Psi$ of ghost number $-1$. We set

$$\Psi = \bar{c}_\mu F_\mu \,, \tag{3.20}$$

where $F_\mu[H]$ is the usual gauge fixing function. We choose it to implement De Donder gauge fixing:

$$F_\mu = \partial_\nu H_{\nu\mu} - \partial_\mu \varphi \,. \tag{3.21}$$

Under the canonical transformation (2.57), *cf.* (2.46), which takes us from the non-minimal gauge invariant basis to the gauge fixed basis (or vice versa), we thus only change $\bar{c}_\mu^*$ and $H_{\mu\nu}^*$. Explicitly we have:[9]

$$\bar{c}_\mu^* \big|_{\mathrm{gf}} = \bar{c}_\mu^* \big|_{\mathrm{gi}} + F_\mu \,, \tag{3.22}$$
$$H_{\mu\nu}^* \big|_{\mathrm{gf}} = H_{\mu\nu}^* \big|_{\mathrm{gi}} - \partial_{(\mu} \bar{c}_{\nu)} + \tfrac{1}{2} \delta_{\mu\nu} \partial \cdot \bar{c} \,.$$

Applying this to (3.7), we get $S_0$ in gauge fixed basis:

$$L_0 = \frac{1}{2} H_{\mu\nu} (\triangle^\Lambda)^{-1}_{\mu\nu,\alpha\beta} H_{\alpha\beta} - \bar{c}_\mu \Box^\Lambda c_\mu - i b_\mu (C^\Lambda)^{-1} F_\mu + \frac{1}{2\alpha} b_\mu (C^\Lambda)^{-1} b_\mu$$
$$- 2 \partial_\mu c_\nu (C^\Lambda)^{-1} H_{\mu\nu}^* - i b_\mu (C^\Lambda)^{-1} \bar{c}_\mu^* \,, \tag{3.23}$$

where $\Box^\Lambda = \Box/C^\Lambda$. This is of the required form (2.48).

In this basis, we can read off the BRST transformation from (2.50) and the Kozsul–Tate differential from (2.51). However recalling that we label these fermionic differentials by their antighost number, we need to recognise that the Kozsul-Tate differential now splits into a piece, $Q$, that leaves the antighost number unchanged and the piece, $Q^-$, that lowers it by one. Therefore we now write (2.51) as:

$$(Q_0 + Q_0^-) \Phi_A^* = (S_0, \Phi_A^*) \,. \tag{3.24}$$

Although we label the (free) antighost neutral piece by $Q_0$ there is no confusion with the BRST transformation $Q_0 \Phi^A$ because the latter acts only on fields, not antifields. Thus we find the following non-vanishing transformations:

$$Q_0 H_{\mu\nu} = 2\partial_{(\mu} c_{\nu)} \,, \quad Q_0 \bar{c}_\mu = i b_\mu \,, \quad Q_0 H_{\mu\nu}^* = i \partial_{(\mu} b_{\nu)} - \frac{i}{2} \delta_{\mu\nu} \partial \cdot b \,, \quad Q_0 \bar{c}_\mu^* = -\Box c_\mu \,,$$
$$Q_0^- H_{\mu\nu}^* = -2 G_{\mu\nu}^0 \,, \quad Q_0^- c_\mu^* = -\Box \bar{c}_\mu - 2\partial_\nu H_{\nu\mu}^* \,, \tag{3.25}$$

where only the first two are BRST transformations, and the rest are Kozsul–Tate differentials. We already know from (A.11), that the measure operators $\Delta^-$ and $\Delta^=$ keep the same form (3.12). Since the map to gauge fixed basis is a quantum canonical transformation, the full BRST charge $s_0$ is still nilpotent, and since we continue to consistently label the pieces by their antighost number, the same arguments as before establish that the charges (3.25) also satisfy the individual nilpotency relations (3.16). On the other hand, by explicit computation we see that the non-vanishing anticommutators (3.19) grow an extra piece:

$$\{Q_0, \Delta^=\} = -\{Q_0^-, \Delta^-\} = 2 \frac{\partial}{\partial c_\nu^*} C^\Lambda \partial_\mu \frac{\partial}{\partial H_{\mu\nu}} + \frac{\partial}{\partial c_\mu^*} C^\Lambda \Box \frac{\partial}{\partial \bar{c}_\mu^*} \,. \tag{3.26}$$

---

[9]defining vector contraction as $u \cdot v = u_\mu v_\mu$.

## 3.3 Comparing the two

Although by the map (3.22), any results derived in this gauge fixed basis are equivalent to any results derived in the gauge invariant basis, we see as advertised that it is simpler when dealing with just the algebra, to work in the gauge invariant basis. Indeed in gauge invariant basis we had only the four non-vanishing transformations (3.2), (3.8), (3.9), whereas in the gauge fixed basis we have the six non-vanishing transformations (3.25), and furthermore the non-vanishing anticommutators (3.26) are more complicated.

When dealing with the BRST cohomology in the gauge invariant basis, we can furthermore specialise to the minimal basis. This means we can drop $\bar{c}_\mu$ and $b_\mu$ and thus (3.8), leading to only three non-vanishing transformations, and also drop $\bar{c}_\mu^*$, leading to a simpler form for $\Delta^-$ in (3.12).

To see this we note that we want to find the non-trivial solutions for $s_0 S_1 = 0$, as in (2.52). If we assume that $S_1$ does not contain $\bar{c}_\mu$, we are not forced to include it, since it does not appear in the $S_0$ given in (3.7), and the QMF, (2.49), contains only field differentials. There is no $b_\mu^*$. Altogether, this means that through the QMF, we never get a modification containing

$$\frac{\partial S_1}{\partial \bar{c}_\mu^*} \quad \text{or} \quad \frac{\partial S_1}{\partial b_\mu}, \tag{3.27}$$

and therefore it does not help in finding non-trivial solutions to include dependence on $\bar{c}_\mu^*$ or $b_\mu$ in $S_1$. Dropping the dependence on $\bar{c}_\mu$, $\bar{c}_\mu^*$ and $b_\mu$, gives us the minimal basis (3.6). As we stated in sec. 2.7 and will illustrate in sec. 7.1, this already contains all the information on how diffeomorphism invariance is realised through the BRST cohomology.

For orders beyond first order, the flow equation will force dependence on the non-minimal fields. However solutions for $S_1$ containing these fields are linearly independent of the solutions in the minimal basis.

## 3.4 Comments on the conformal mode instability and other signs

The central observation that leads to the new quantisation is that the gravitational action (3.3) is unbounded from below. This is a gauge invariant statement: the unboundedness is caused by the fact that the functional integral will explore arbitrarily high scalar curvature, with positive scalar curvature being the problematic case. In sec. 7 we will instead be exploring a more general space of interactions built on the free graviton action (3.1). However the free graviton action also has these problems. Indeed, splitting the graviton field $H_{\mu\nu}$ into its $SO(d)$ irreducible parts:

$$H_{\mu\nu} = h_{\mu\nu} + \frac{2}{d}\varphi\,\delta_{\mu\nu} \tag{3.28}$$

(thus $h_\mu{}^\mu = 0$ is traceless), and considering purely traceful perturbations $\varphi$ (i.e. setting $h_{\mu\nu} = 0$) we see that the action is unbounded below for these modes (for $d > 8/5$).

The situation is obscured by linearised gauge invariance. Using (3.11), the gauge invariant statement is that the action is unbounded below in the following direction

$$2\varphi - \frac{\partial_{\alpha\beta}^2}{\Box}H_{\alpha\beta} = 2\left(1 - \frac{1}{d}\right)\varphi - \frac{\partial_{\alpha\beta}^2}{\Box}h_{\alpha\beta} = \frac{1}{-\Box}R^{(1)}. \tag{3.29}$$

Using different gauge choices we can shift the instability to different modes, but we cannot remove it. Indeed in the Landau gauge limit of the De Donder gauge (3.21), where we insist that $F_\mu = 0$ identically, the conformal mode coincides with this (linearised) gauge invariant quantity:

$$\varphi = \frac{1}{-\Box}R^{(1)}. \tag{3.30}$$

Completing the square for the $b_\mu$ auxiliary field and integrating it out, gives for just the graviton part, the gauge fixed Lagrangian $\mathcal{L}_0 = \bar{\mathcal{L}}_0$, where

$$\bar{\mathcal{L}}_0 = \frac{1}{2} H_{\mu\nu} (\triangle^\Lambda)^{-1}_{\mu\nu,\alpha\beta} H_{\alpha\beta} + \frac{\alpha}{2} F_\mu (C^\Lambda)^{-1} F_\mu. \tag{3.31}$$

Again splitting the graviton field $H_{\mu\nu}$ into its $SO(d)$ irreducible parts, and choosing $\alpha = 2$ gauge and $d = 4$ dimensions, gives the action that was used to derive the eigenoperators (1.4) in ref. [5]:

$$\bar{\mathcal{L}}_0 = -\frac{1}{2} h_{\mu\nu} \square^\Lambda h_{\mu\nu} + \frac{1}{2} \varphi \square^\Lambda \varphi. \tag{3.32}$$

It has the advantage that at the free level, the instability is then isolated in the $\varphi$ sector. We will not integrate out $b_\mu$ however since that would leave us only with on-shell BRST. Instead we will keep the auxiliary field and thus ensure that the nilpotency relations (3.16) remain valid off shell.

It is possible to consider slightly more general linearised transformations than (3.2). Indeed in quantising the Einsten-Hilbert action one can choose to treat not the metric but the density

$$|g|^{w/2} g_{\mu\nu} = \delta_{\mu\nu} + \kappa H_{\mu\nu}, \tag{3.33}$$

where use of the determinant $|g|$ means that the left hand side is a tensor density of weight $-w$. This implies that $H_{\mu\nu}$ transforms at the free level with an extra piece:

$$Q_0 H_{\mu\nu} = \partial_\mu c_\nu + \partial_\nu c_\mu + w \delta_{\mu\nu} \partial \cdot c. \tag{3.34}$$

Clearly this leaves the transformation of $h_{\mu\nu}$ alone, affecting only the scalar component:

$$Q_0 \varphi = \left(1 + \frac{d}{2} w\right) \partial \cdot c. \tag{3.35}$$

In fact taking the determinant of (3.33) we have to order $\kappa$

$$|g| = 1 + \frac{4\kappa}{2 + wd} \varphi, \tag{3.36}$$

so at the free level the change of variables amounts to simply rescaling $\varphi$, and for this reason we will not consider it further.

However we do need to comment on the exceptional case, $w = -2/d$. In this case, taking the determinant of (3.33) shows that $\varphi$ vanishes identically. This is unimodular gravity [69, 70], where the metric $\hat{g}_{\mu\nu}$ is constrained to have unit determinant. It can be treated by writing $\hat{g}_{\mu\nu} = g_{\mu\nu}/|g|^{1/d}$ [71, 72], which is precisely the left hand side of (3.33) in this case. It is still the case that the Euclidean signature Einstein-Hilbert action (3.3) is unbounded below to arbitrarily large positive curvature, but the problem is now confined to the interactions. We expect that this must still leave its mark on the Wilsonian RG, but clearly the analysis is now much more involved. Similar comments also apply to the variants of first order formalism where the connection (or spin connection in Cartan formalism) is treated as fundamental. Again the instability to large positive curvature is still present but its consequences for the Wilsonian RG are not so straightforward to analyse.

As we have just seen (and see also sec. 3.5), it is not possible to get rid of the instability, except by the expedient of rotating to complex metrics.[10] Nevertheless since the new quantisation all hangs on a sign, it behoves us to be more than usually careful with signs elsewhere. Following [26] it is common to be careless with signs when auxiliary fields, here $b_\mu$, are introduced. This is harmless since in standard treatments the auxiliary field only ever appears

---

[10]In fact a purely imaginary $\varphi \mapsto i\varphi$ [7] cannot be maintained if diffeomorphism invariance, $\delta\varphi = \nabla\cdot\xi$, is also to be respected.

in the action up to quadratic level, and indeed can be integrated out as we have just seen. Then following [26], in preference to introducing $i$ into the action (3.7) one can work without the explicit $i$, by rotating $b_\mu \mapsto -i\, b_\mu$, at the expense of having the wrong sign for the $\sim b_\mu^2$ term in (3.7). Note that after this rotation the action is still not real however, unless one goes the extra step and formally treats the Grassmann even pair $\{c_\mu^*, \bar{c}_\mu^*\}$, and Grassmann odd pair $\{c_\mu, \bar{c}_\mu\}$, as real variables.

In fact the $i$ is there for good reasons. To see this most clearly we follow refs. [26, 27], *i.e.* set $C^\Lambda = 1$ and replace the antifields by $\partial_r \Psi / \partial \Phi^A$. The latter can be done here simply by setting the antifields to zero in gauge fixed basis. Then in the Landau gauge limit, $\alpha \to \infty$, we see from (3.23) that the functional integral over $b_\mu$ is a functional Fourier transform expressing the fact that we have inserted into the partition function (2.3) the functional delta function $\delta[F_\mu]$, as it should be. If we had not put the $i$ into (3.7) we would at this point have to do so, *e.g.* by rotating $b_\mu \mapsto i\, b_\mu$, in order to turn an otherwise divergent integral over $\exp b_\mu F_\mu$, into the Fourier transform.

## 3.5 Propagators

In order to find the eigenoperators, we need to solve the equation (2.54) by separation of variables. As we can see in (2.22), (2.20), this requires knowing the propagators. Although $\triangle_{H_{\mu\nu} H_{\alpha\beta}}^{-1}$ is not itself invertible, as part of the larger matrix $\triangle_{AB}^{-1}$ defined via (3.23), it is. As discussed in the previous section, it is clear however that $\triangle_{AB}^{-1}$ is not positive definite (for any choice of $\alpha$). Indeed setting all fields to zero except again $H_{\mu\nu} = \frac{2}{d}\varphi\,\delta_{\mu\nu}$, the Lagrangian (3.23) is unbounded from below.

The matrix $\triangle_{AB}^{-1}$ in $H_{\mu\nu}, b_\mu$ space, can be inverted by *e.g.* using a transverse traceless decomposition:

$$h_{\mu\nu} = 2\partial_{(\mu}\xi_{\nu)}^T + 2\left(\partial_{\mu\nu}^2\xi - \frac{1}{d}\delta_{\mu\nu}\Box\xi\right) + h_{\mu\nu}^T, \quad b_\mu = b_\mu^T + \partial_\mu b, \tag{3.37}$$

where $h_{\mu\mu}^T = 0$, the generator of linearised diffeomorphisms is split into transverse and longitudinal parts $\xi_\mu = \xi_\mu^T + \partial_\mu\xi$, similarly $b_\mu = b_\mu^T + \partial_\mu b$, and the transverse fields satisfy $\partial_\mu h_{\mu\nu}^T = 0$ *etc.* It is also helpful to absorb $\Box\xi$ into $\varphi$ by defining $\varphi' = \varphi - \Box\xi$ in (3.28). Anyway, noting that

$$\triangle^{AB} = \langle \Phi^A \Phi^B \rangle, \tag{3.38}$$

and writing

$$\Phi^A(x) = \int \frac{d^d p}{(2\pi)^d}\, e^{-ip\cdot x}\, \Phi^A(p), \tag{3.39}$$

we find the following propagators:

$$\langle H_{\mu\nu}(p) H_{\alpha\beta}(-p)\rangle = \frac{\delta_{\mu(\alpha}\delta_{\beta)\nu}}{p^2} + \left(\frac{4}{\alpha} - 2\right)\frac{p_{(\mu}\delta_{\nu)(\alpha}p_{\beta)}}{p^4} - \frac{1}{d-2}\frac{\delta_{\mu\nu}\delta_{\alpha\beta}}{p^2}, \tag{3.40}$$

$$\langle b_\mu(p) H_{\alpha\beta}(-p)\rangle = -\langle H_{\alpha\beta}(p) b_\mu(-p)\rangle = 2\,\delta_{\mu(\alpha}p_{\beta)}/p^2, \tag{3.41}$$

$$\langle b_\mu(p) b_\nu(-p)\rangle = 0, \tag{3.42}$$

$$\langle c_\mu(p)\bar{c}_\nu(-p)\rangle = -\langle \bar{c}_\mu(p) c_\nu(-p)\rangle = \delta_{\mu\nu}/p^2. \tag{3.43}$$

Note that the $b_\mu$ does not actually propagate into itself. Projecting the top line into its irreducible representations gives:

$$\begin{aligned}
\langle h_{\mu\nu}(p) h_{\alpha\beta}(-p)\rangle =&\ \frac{\delta_{\mu(\alpha}\delta_{\beta)\nu}}{p^2} + \left(\frac{4}{\alpha} - 2\right)\frac{p_{(\mu}\delta_{\nu)(\alpha}p_{\beta)}}{p^4} + \frac{1}{d^2}\left(\frac{4}{\alpha} - d - 2\right)\frac{\delta_{\mu\nu}\delta_{\alpha\beta}}{p^2} \\
&+ \frac{2}{d}\left(1 - \frac{2}{\alpha}\right)\frac{\delta_{\alpha\beta}p_\mu p_\nu + p_\alpha p_\beta \delta_{\mu\nu}}{p^4},
\end{aligned} \tag{3.44}$$

and

$$\langle h_{\mu\nu}(p)\,\varphi(-p)\rangle = \langle \varphi(p)\,h_{\mu\nu}(-p)\rangle = \left(1 - \frac{2}{\alpha}\right)\left(\frac{\delta_{\mu\nu}}{d} - \frac{p_\mu p_\nu}{p^2}\right)\frac{1}{p^2}, \tag{3.45}$$

$$\langle \varphi(p)\,\varphi(-p)\rangle = \left(\frac{1}{\alpha} - \frac{d-1}{d-2}\right)\frac{1}{p^2}. \tag{3.46}$$

For future calculations in this theory, it will undoubtedly be helpful to confirm that physical results are independent of the choice of gauge parameter $\alpha$. However just as we saw in sec. 3.4, the choice $\alpha = 2$ leads to simplifications. Indeed in this case $h_{\mu\nu}$ and $\varphi$ propagate separately. In this paper we will from here on specialise to $\alpha = 2$. Now in the $H_{\mu\nu}$ sector we have:

$$\langle H_{\mu\nu}(p)\,H_{\alpha\beta}(-p)\rangle = \frac{\delta_{\mu(\alpha}\delta_{\beta)\nu}}{p^2} - \frac{1}{d-2}\frac{\delta_{\mu\nu}\delta_{\alpha\beta}}{p^2}, \tag{3.47}$$

$$\langle h_{\mu\nu}(p)\,h_{\alpha\beta}(-p)\rangle = \frac{\delta_{\mu(\alpha}\delta_{\beta)\nu} - \frac{1}{d}\delta_{\mu\nu}\delta_{\alpha\beta}}{p^2}, \tag{3.48}$$

$$\langle h_{\mu\nu}(p)\,\varphi(-p)\rangle = \langle \varphi(p)\,h_{\mu\nu}(-p)\rangle = 0, \tag{3.49}$$

$$\langle \varphi(p)\,\varphi(-p)\rangle = -\frac{d}{2(d-2)}\frac{1}{p^2}. \tag{3.50}$$

We note that $h_{\mu\nu}$ propagates with the right sign, and that the numerator is just the projector onto traceless tensors. We note that $\varphi$ propagates with wrong sign for all $d > 2$.

# 4 Eigenoperators

As in ref. [5], the Hilbert space $\mathfrak{L}$ of bare interactions is constructed from those eigenoperators that form an orthonormal basis about the Gaussian fixed point, using the natural measure. This measure is the Sturm-Liouville weight function for the corresponding eigenoperator equation. As shown in ref. [5], the eigenoperators follow from those derived for the non-derivative interactions. Therefore we derive these first.

The eigenoperator equation follows from (2.54), where the tadpole integral $a_1$ is defined in (2.22) and (2.20). Recognising that

$$\frac{\partial}{\partial H_{\alpha\beta}} = \frac{\partial h_{\mu\nu}}{\partial H_{\alpha\beta}}\frac{\partial}{\partial h_{\mu\nu}} + \frac{\partial \varphi}{\partial H_{\alpha\beta}}\frac{\partial}{\partial \varphi} = \frac{\partial}{\partial h_{\alpha\beta}} + \frac{1}{2}\delta_{\alpha\beta}\frac{\partial}{\partial \varphi}, \tag{4.1}$$

and using the propagators in the last subsection, we can write out (2.54) in full:

$$\dot{S}_1 = -\frac{1}{2}\int \frac{d^d p}{(2\pi)^d}\frac{\dot{C}^\Lambda(p)}{p^2}\left\{\frac{\delta^2}{\delta h_{\mu\nu}(p)\,\delta h_{\mu\nu}(-p)} - \frac{d}{2(d-2)}\frac{\delta^2}{\delta\varphi(p)\,\delta\varphi(-p)}\right.$$
$$\left. + 2p_\mu\frac{\delta^2}{\delta H_{\mu\nu}(p)\,\delta b_\nu(-p)} - 2\frac{\delta_l}{\delta\bar{c}_\mu(p)}\frac{\delta_r}{\delta c_\mu(-p)}\right\}S_1. \tag{4.2}$$

From here on in this subsection we abandon DeWitt notation. The reader will note that we split $H_{\alpha\beta}$ into its irreducible representations only where it is illuminating to do so. The choice of left and right derivative is made for the ghosts to make explicit the minus sign that one gets for a ghost closed loop.

## 4.1 Non-derivative eigenoperators

To get the eigenoperator equation for non-derivative interactions we set:

$$S_1 = \int d^d x \, V(\Phi, \Phi^*, \Lambda), \tag{4.3}$$

for some 'potential' $V$, work in dimensionless variables $\{\tilde{\Phi}, \tilde{\Phi}^*\}$ constructed using their engineering dimensions (listed in table 1, this being the scaling dimension at the Gaussian fixed point), write similarly $V = \Lambda^d \tilde{V}$, and separate variables:

$$\tilde{V}_\Lambda(\tilde{\Phi}, \tilde{\Phi}^*) = e^{\lambda t} \, \tilde{V}(\tilde{\Phi}, \tilde{\Phi}^*). \tag{4.4}$$

Notice that the $H-b$ cross-term vanishes for non-derivative interactions, and thus neither $b_\mu$ nor the antifields enter explicitly in the eigenoperator equation. Their presence will just be felt via their dimensions, as we show later. Dropping these variables for the moment we see therefore that the eigenoperator equation further separates into parts, one for each $h_{\mu\nu}$, for $\varphi$, and for each pair $\{\bar{c}_\mu, c_\mu\}$.

### 4.1.1 Conformal factor eigenoperators

We start with $\varphi$ since this will set our conventions when $d \neq 4$. The double-functional derivative in (4.2) just computes the tadpole integral already introduced in (1.1):

$$\Omega_\Lambda = |\langle \varphi(x)\varphi(x)\rangle| = \frac{d}{2(d-2)} \int \frac{d^d p}{(2\pi)^d} \frac{C^\Lambda(p)}{p^2}. \tag{4.5}$$

Recalling that $C^\Lambda(p) = C(p^2/\Lambda^2)$ for some function $C$, we have that $\Omega_\Lambda = \Lambda^{d-2}/(2a^2)$, where the non-universal constant $a > 0$ is thus given by

$$\frac{1}{a^2} = \frac{d}{d-2} \int \frac{d^d \tilde{p}}{(2\pi)^d} \frac{C(\tilde{p}^2)}{\tilde{p}^2}. \tag{4.6}$$

These definitions coincide with those in ref. [5] when $d = 4$. The eigenoperator equation is then:

$$-\lambda \tilde{V}(\tilde{\varphi}) - \left(\frac{d-2}{2}\right) \tilde{\varphi} \, \tilde{V}' + d \, \tilde{V} = \left(\frac{d-2}{4a^2}\right) \tilde{V}'', \tag{4.7}$$

where prime is differentiation with respect to the argument. Multiplying through by $2/(d-2)$, we see that it then coincides the $d = 4$ equation:

$$-\lambda_4 \tilde{V}(\tilde{\varphi}) - \tilde{\varphi} \, \tilde{V}' + 4\tilde{V} = \frac{\tilde{V}''}{2a^2}, \tag{4.8}$$

if we also redefine $\lambda$:

$$\lambda_4 = \frac{2}{d-2}(\lambda + d - 4). \tag{4.9}$$

It is therefore the same Sturm-Liouville eigenfunction equation analysed in detail in ref. [5], and we can read off its properties from there. In particular its weight function is still given by (1.1), which in scaled variables is simply

$$e^{+a^2 \tilde{\varphi}^2}. \tag{4.10}$$

Functions that are square integrable under this measure form the Hilbert space $\mathfrak{L}_-$. The eigenoperators

$$\delta_n(\tilde{\varphi}) = \frac{a}{\sqrt{\pi}} \frac{\partial^n}{\partial \tilde{\varphi}^n} e^{-a^2 \tilde{\varphi}^2}, \tag{4.11}$$

(integer $n \geq 0$) are orthonormal under this weight:

$$\int_{-\infty}^{\infty} d\tilde{\varphi} \; e^{a^2 \tilde{\varphi}^2} \delta_n(\tilde{\varphi}) \delta_m(\tilde{\varphi}) = \frac{a}{\sqrt{\pi}} \left(2a^2\right)^n n! \, \delta_{nm}, \tag{4.12}$$

and span the space. From $\lambda_4 = 5 + n$, we have $\lambda = \frac{3d}{2} - 1 + \left(\frac{d-2}{2}\right)n$, and thus their scaling dimension is

$$[\delta_n] = d - \lambda = -\left(\frac{d-2}{2}\right)(n+1). \tag{4.13}$$

In dimensionful terms they are the operators listed in (1.2), where we see that their scaling dimension continues to be also their engineering dimension.

Interactions that lie in $\mathfrak{L}_-$ are characterised by having a decay for large amplitude faster than $e^{-a^2 \tilde{\varphi}^2/2}$. In ref. [5], it was shown that small perturbations in $\mathfrak{L}_-$, stay in $\mathfrak{L}_-$ under the RG flow as $\Lambda$ is increased. We will review this in sec. 6. This is why we interpret restricting to $\mathfrak{L}_-$ as a quantisation condition on the bare interactions. General interactions outside $\mathfrak{L}_-$ can also be mooted, but then there is no natural sense in which these are related to an expansion over some set of distinguished operators. Without this, the RG itself breaks down, since then it is no longer clear how to split an arbitrary perturbation into its relevant and irrelevant parts. Note that bare operators that are excluded by this quantisation condition include any polynomial interaction, and in particular the unit operator. When we embed this structure into the full theory, this will force all bare operators to depend on $\varphi$ [5].

### 4.1.2 Traceless mode eigenoperators

Next we choose a $\mu$ and a $\nu$ and consider a potential made from the one component, $h \equiv h_{\mu\nu}$. Of course this means abandoning SO($d$) invariance but we do so only temporarily. If we define

$$\frac{1}{a_h^2} = 2\chi \int \frac{d^d \tilde{p}}{(2\pi)^d} \frac{C(\tilde{p}^2)}{\tilde{p}^2}, \tag{4.14}$$

where $\chi = 2/(1 + \delta_{\mu\nu})$ takes into account that $h_{\mu \neq \nu}$ appears twice on the right hand side of (4.2), then by comparison with $\tilde{\varphi}$ we see that we obtain the following eigenequation:

$$-\lambda \tilde{V}(\tilde{h}) - \left(\frac{d-2}{2}\right) \tilde{h} \tilde{V}' + d \tilde{V} = -\left(\frac{d-2}{4a_h^2}\right) \tilde{V}''. \tag{4.15}$$

Multiplying through by $2/(d-2)$ this coincides with the eigenoperator equation for a scalar field in four dimensions with the correct sign for its kinetic term:

$$-\lambda_4 \tilde{V}(\tilde{h}) - \tilde{h} \tilde{V}' + 4\tilde{V} = -\frac{\tilde{V}''}{2a_h^2}, \tag{4.16}$$

where $\lambda_4$ is again (4.9). The change of sign on the right hand side is crucial. Again reading off from ref. [5] (see also [73–75]), the Sturm-Liouville weight is now $e^{-a_h^2 \tilde{h}^2}$. Perturbations that are square integrable under this weight form a Hilbert space $\mathfrak{L}_+$. The eigenoperators

$$\mathcal{O}_n(\tilde{h}) = H_n(a_h \tilde{h})/(2a_h)^n = \tilde{h}^n - n(n-1)\tilde{h}^{n-2}/4a_h^2 + \cdots, \tag{4.17}$$

with $\lambda_4 = 4 - n$ and $n$ a non-negative integer, and $H_n$ the $n^{\text{th}}$ Hermite polynomial, form a complete orthornormal basis for this space. The scaling dimension for these operators

$$[\mathcal{O}_n] = d - \lambda = \left(\frac{d-2}{2}\right)n, \tag{4.18}$$

is just the engineering dimension $[h^n]$ of the highest power in (4.17). This reflects the fact that the quantum part of (4.15) (the right hand side) does not contribute to the highest power, but simply adds the tadpole corrections, which are the lower powers appearing in (4.17).

From (4.2), a non-derivative eigenoperator $\tilde{V}(\tilde{h}_{\mu\nu})$, now considering all the $h_{\mu\nu}$ components together, is thus given by a sum over products

$$\sum_j a_j \prod_k \mathcal{O}_{n_j^k}(\tilde{h}_{\mu_j^k \nu_j^k}),\tag{4.19}$$

with $\Lambda$ independent coefficients $a_j$, such that the highest power $\sum_k n_j^k = n$ is the same in each product.

We will shortly consider these operators as parts of a larger operator. As such, the non-derivative part which we are now studying, need not be Lorentz invariant on its own, but will need to be Lorentz covariant. Since we started with an SO($d$) invariant equation (4.2), we are guaranteed that by suitable choice of the $a_j$ we can recover the correct Lorentz covariant structure. In fact we have just seen that the right hand side of (4.2) only generates tadpole corrections, so we know already that these eigenoperators are given by the independent monomials $h_{\mu\nu}^n$ with the right covariance, together with lower powers generated by the tadpole correction on the right hand side of (4.2). Since for fixed $n$, such operators span a finite dimensional space we can furthermore use Gram-Schmidt orthogonalisation to choose them to be orthonormal under the combined Sturm-Liouville weight:

$$\exp - \left\{ \sum_\mu a_h^2 \tilde{h}_{\mu\mu}^2 + \sum_{\mu<\nu} a_h^2 \tilde{h}_{\mu\nu}^2 \right\} = \exp - \frac{d\, a^2}{2(d-2)} \tilde{h}_{\mu\nu}^2,\tag{4.20}$$

where on the right hand side we compare (4.14) and (4.6), recall that $\chi = 2$ for $\mu \neq \nu$, and reinstate Einstein summation convention.

However we will not insist that they are orthogonal. The properties we need are that the pure $h_{\mu\nu}$ non-derivative eigenoperators are polynomials whose scaling dimension is given by the engineering dimension of their highest power, and that these operators span the space of functions square integrable under (4.20).

### 4.1.3 Ghost eigenoperators

Now consider a 'potential' for just the ghosts. In order to keep the notation readable, we will drop the tildes: all quantities will however be scaled. Using (4.6), we get from (4.2) the eigenoperator equation:

$$-\lambda V(\bar{c}, c) - \frac{d-2}{2}\left( \bar{c}_\mu \frac{\partial_l}{\partial \bar{c}_\mu} + c_\mu \frac{\partial_l}{\partial c_\mu} \right)V + d\, V = \frac{(d-2)^2}{d\, a^2} \frac{\partial_l}{\partial \bar{c}_\mu} \frac{\partial_r}{\partial c_\mu} V.\tag{4.21}$$

Since the ghosts are Grassmann, $V$ can only be a polynomial. The right hand side cannot contribute to a top term (the highest power of $c_\mu$ or $\bar{c}_\mu$ in this polynomial), which must thus satisfy the left hand side alone. Let the top term of an eigenoperator solution $\mathcal{O}(\bar{c}, c)$ contain $n$ factors of $c$ and $\bar{n}$ factors of $\bar{c}$. Then $\lambda = d - (n+\bar{n})(d-2)/2$. Thus we see that the scaling dimension for the operator, $d - \lambda = (n+\bar{n})(d-2)/2$, is just the engineering dimension of this top term, just like for $h_{\mu\nu}$. We can cast (4.21) in Sturm-Liouville form and thus show that the Sturm-Liouville weight function is

$$\exp\left( -\frac{d\, a^2}{d-2} \bar{c}_\mu c_\mu \right),\tag{4.22}$$

under which the eigenoperators are orthogonal in the following sense.

We now choose a $\mu$ and consider the operators made from just the two components $\bar{c} \equiv \bar{c}_\mu$ and $c \equiv c_\mu$. The entire set is

$$\mathcal{O}_0 = 1, \quad \mathcal{O}_1 = c, \quad \mathcal{O}_{\bar{1}} = \bar{c}, \quad \text{and} \quad \mathcal{O}_2 = \bar{c}c + \frac{d-2}{da^2}, \tag{4.23}$$

since higher powers vanish by the Grassmann property. These are orthonormal under (4.22) in the sense that

$$\int d\bar{c}\, dc \; \mathrm{e}^{\frac{da^2}{d-2}c\bar{c}} \mathcal{O}_a \mathcal{O}_b = \eta_{ab}, \tag{4.24}$$

where $\eta_{ab} = 0$ for all combinations apart from

$$\eta_{00} = \frac{d\, a^2}{d-2}, \quad \eta_{1\bar{1}} = -\eta_{\bar{1}1} = 1, \quad \text{and} \quad \eta_{22} = -\frac{d-2}{d\, a^2}. \tag{4.25}$$

Again we will consider these operators as parts of a larger operator, so they only need to be Lorentz covariant. Again, since (4.21) is Lorentz invariant we are guaranteed to be able to build these operators, and they are just given by starting with a top term of the right Lorentz covariance.

### 4.1.4 Auxiliary field and/or antifield eigenoperators

Since non-derivative operators containing only $b_\mu$ and/or the $\Phi_A^*$ are not affected by the right hand side of (4.2), in scaled variables the eigenoperator equation is just (using table 1):

$$-\lambda \tilde{V}(\tilde{b}, \tilde{\Phi}^*) - \frac{d}{2}\left( \tilde{b}_\mu \frac{\partial}{\partial \tilde{b}_\mu} + \tilde{\Phi}_A^* \frac{\partial_l}{\partial \tilde{\Phi}_A^*} \right) \tilde{V} + d\tilde{V} = 0, \tag{4.26}$$

which just says that $d - \lambda$ is the engineering dimension of this combination. Remembering that we require only Lorentz covariance, if we consider purely $b$ interactions, the only sensible choice is thus a Lorentz covariant monomial. Including also the $\Phi_A^*$, the non-derivative eigenoperators are just the linearly independent Lorentz covariant monomials, and their scaling dimensions are just their engineering dimensions. On the other hand (4.26) is not of Sturm-Liouville type: there are no natural orthonormality or Hilbert space properties inherited for parts of interactions depending on these fields. Instead they must be specified as we have just done. For each specification, the rest of the non-derivative operator, namely the eigenoperator parts in $H_{\mu\nu}$ and the ghosts, do form a Hilbert space of interactions as we have seen.

## 4.2 General eigenoperators

Since the eigenoperator equation (4.2) for a non-derivative eigenoperator, separates into the parts we have just considered, we know that the general non-derivative eigenoperator just corresponds to the linearly independent Lorentz invariant sums of products of these eigenoperators with the same overall scaling dimension. The overall scaling dimension is given by the sum of its parts, namely the engineering dimension for the $\delta_\Lambda^{(n)}(\varphi)$ operators (4.13), the engineering dimension (4.18) for the top term in the $h_{\mu\nu}$ polynomials, the engineering dimension for the top term in the ghost polynomials, and the engineering dimension for $b_\mu$ and $\Phi_A^*$ pieces.

Now consider what happens when we include also derivative interactions in the eigenoperator. The behaviour is similar to the $h$ and ghost polynomial solutions above, because if the tadpole operator, *i.e.* the right hand side of (4.2), hits a field that has spacetime derivatives, the result is an interaction where this piece is eliminated. For example a piece $\partial_\eta h_{\alpha\beta}\, \partial_\lambda h_{\mu\nu}$

can be eliminated by the $h$ functional derivatives, being replaced by a coefficient proportional to

$$\delta_{\eta\lambda}\left(\delta_{\mu(\alpha}\delta_{\beta)\nu} - \frac{1}{d}\delta_{\mu\nu}\delta_{\alpha\beta}\right)\int\frac{d^d p}{(2\pi)^d}\,\dot{C}^\Lambda(p).\tag{4.27}$$

The result is thus a top term plus tadpole corrections containing less components. Similarly the $H$–$b$ cross-term in the tadpole operator can create tadpole corrections by eliminating for example a $\partial_\mu b_\nu$ piece together with an $h_{\alpha\beta}$, or together with a $\varphi$-differential of the non-derivative $\varphi$ dependence.

The RG eigenvalue therefore involves the tadpole operator only when both $\varphi$ derivatives hit the coefficient function $f_\Lambda(\varphi)$. We thus find that the non-derivative part is made up of the eigenoperators for each field that we have already discussed. The left hand side of the eigenoperator equation (4.2), when written in scaled variables, continues to count the total engineering dimension of the top term. In particular the dimension of the space-time derivatives enters via the consistent assignment of engineering dimension for $f$. (For an example worked out in detail, namely for $K(\varphi)(\partial\varphi)^2$, see ref. [5].) We thus see that the general eigenoperator can be written in the form:

$$\delta_\Lambda^{(n)}(\varphi)\,\sigma(\partial,\partial\varphi,h,\bar{c},c,b,\Phi^*)+\cdots.\tag{4.28}$$

We have displayed the 'top term'. $\sigma$ is a Lorentz invariant monomial involving some or all of the components indicated, in particular the arguments $\partial\varphi, h, \bar{c}, c, b, \Phi^*$ can appear as they are, or differentiated any number of times. The tadpole operator then in particular generates tadpole corrections involving less fields in $\sigma$. These are the terms we indicate with the ellipses. Let $d_\sigma = [\sigma]$ be the engineering dimension of the monomial $\sigma$, then the scaling dimension $d - \lambda$ of the corresponding eigenoperator is just the engineering dimension $D_\sigma = d_\sigma + [\delta_n]$, where $[\delta_n]$ was defined in eqn. (4.13).

Note that the result (4.28) is the appropriate generalisation of the result (1.4) found in ref. [5], to the case where the quantum realisation of diffeomorphism invariance can now be addressed. In particular the scaling dimension $D_\sigma$ is given by the same formula mentioned in the Introduction.

## 5 The Hilbert space of bare interactions

For a given choice of monomial for the differentiated fields and undifferentiated $b_\mu$ and $\Phi_A^*$, the remaining parts of the eigenoperator thus factorise into pieces that have orthonormality properties under the appropriate measure, namely (4.10), (4.20), or (4.22). Bringing together these factors, the undifferentiated ghost and $H_{\mu\nu}$ parts form coefficient-eigenoperators which span $\mathfrak{L}$, the Hilbert space of bare non-derivative interactions in these fields that are square integrable under the combined amplitude measure:

$$\mu = \exp\frac{1}{2\Omega_\Lambda}\left(\varphi^2 - \frac{d}{2(d-2)}h_{\mu\nu}^2 - \frac{d}{d-2}\bar{c}_\mu c_\mu\right)\tag{5.1}$$

(now writing the measure in dimensionful terms). Note that this formula is the appropriate generalisation of the one found in ref. [5], quoted in (1.3), to general dimension $d$ and to the case where quantum realisations of diffeomorphism invariance can be studied. One should really think of the interactions as furnishing many realisations of $\mathfrak{L}$, one for each choice of monomial of $b_\mu$, $\Phi_A^*$, and the differentiated fields. There is no sense in which eigenoperators with different dependence on $b_\mu$, $\Phi_A^*$, and the differentiated fields, can be directly compared by using the Hilbert space inner product. A related comment is the following. Recall that

these are integrated operators. By integration by parts, we can change the form of the co-efficient function. For example we have that $\delta_n(\tilde{\varphi})\tilde{\Box}\tilde{\varphi}$ and $-(\tilde{\partial}_\mu\tilde{\varphi})^2\delta_{n+1}(\tilde{\varphi})$ are equivalent representations of the same operator. This is not in conflict with the fact that $\delta_n$ and $\delta_{n+1}$ are orthogonal, since they belong to different realisations of $\mathfrak{L}_-$. Although we can change the realisation of $\mathfrak{L}$ this way, it is not possible to map out of $\mathfrak{L}$ by integration by parts. Finally note that one should of course choose a basis of top terms that are independent under integration by parts [5].

We have derived the amplitude measure in gauge fixed basis. However the map to gauge invariant basis (3.22), only changes the dependence on differentiated $H_{\alpha\beta}$ and $\bar{c}_\alpha$ fields. It therefore leaves the measure (5.1) alone and does not change the orthonormality properties of the eigenoperators. The measure is compatible with BRST invariance in the following sense. The BRST transformations (3.2), (3.8), (3.9), in the gauge invariant basis, or (3.25) in the gauge fixed basis, only map to $b_\mu$ or derivative terms. Thus the action of BRST is to move the operator to a different realisation of $\mathfrak{L}$, in the sense explained above.

As already mentioned and reviewed below, the requirement that the bare interactions lie in $\mathfrak{L}$ amounts to a quantisation condition. As in ref. [5], we note that it is preserved term by term when we consider quantum corrections. Thus this will involve differentiating the operators with respect to the field amplitudes, but since this clearly maps polynomials to polynomials, and from (1.2), $\partial_\varphi\delta_\Lambda^{(n)}(\varphi) = \delta_\Lambda^{(n+1)}(\varphi)$, this leaves these operators in $\mathfrak{L}$. Similarly multiplying the eigenoperators by field amplitudes produces an operator that is still in $\mathfrak{L}$, since clearly this again maps polynomials to polynomials, while for $\varphi$ we have [5]:

$$\varphi\,\delta_\Lambda^{(n)}(\varphi) = -n\,\delta_\Lambda^{(n-1)}(\varphi) - \Omega_\Lambda\,\delta_\Lambda^{(n+1)}(\varphi)\,. \tag{5.2}$$

Finally products of the eigenoperators, produce operators that are still in $\mathfrak{L}$. In particular we have [5]:

$$\delta_m(\tilde{\varphi})\,\delta_n(\tilde{\varphi}) = \sum_{j=0}^{\infty}\mathring{c}_{mn}^{j}\,\delta_j(\tilde{\varphi})\,, \tag{5.3}$$

where

$$\mathring{c}_{mn}^{j} = \frac{2^{s-j}a^{2s-2j}}{2\pi^2 j!}\Gamma(s-j)\Gamma(s-m)\Gamma(s-n)\,\delta_{j+m+n=\text{even}}\,, \quad \text{and} \quad 2s = j+m+n+1\,. \tag{5.4}$$

There is one subtlety we have to address however when considering in what sense operators are considered trivial in the BRST cohomology. Recall that such operators $sK$ give vanishing correlators (2.17). If $K$ is quasi-local this is just a change of variables that does not change the physics, and we must discard it. Consider the free cohomology and a candidate interaction in $S$ with a maximum antighost number part $\mathcal{O}^n$ in (3.14). Such a piece should be closed under $Q_0$. We now choose some monomial $\sigma$ that is, without utilising integration by parts, closed under $Q_0$ but not exact. For concreteness (and later), let us spell out what this looks like. In this case $\sigma$ must be made up only of the invariants (3.11) and antifields, differentiated as many times as we wish, and factors of $c_\alpha$ and $\partial_{[\mu}c_{\nu]}$ [22] (that are not further differentiated). To see this, note that the symmetrized derivative $\partial_{(\alpha}c_{\beta)}$ is $Q_0$-exact by (3.2), and a ghost differentiated more than once is also $Q_0$-exact:

$$\partial_{\mu\nu}^2 c^\alpha = Q_0\,\Gamma^{(1)\alpha}_{\phantom{(1)\alpha}\mu\nu}\,, \tag{5.5}$$

where we have introduced the linearised version of the usual connection:

$$\Gamma^{(1)\alpha}_{\phantom{(1)\alpha}\mu\nu} = \tfrac{1}{2}\left(\partial_\mu H_{\alpha\nu} + \partial_\nu H_{\alpha\mu} - \partial_\alpha H_{\mu\nu}\right)\,. \tag{5.6}$$

We thus write this $\sigma$ as:

$$\sigma(c_\alpha, \partial_{[\mu} c_{\nu]}, \partial, \Phi^*, R^{(1)}). \tag{5.7}$$

Now a piece

$$\delta_\Lambda^{(0)}(\varphi) \, \partial \cdot c \, \sigma(c_\alpha, \partial_{[\mu} c_{\nu]}, \partial, \Phi^*, R^{(1)}) = Q_0 \, \delta_\Lambda^{(-1)}(\varphi) \, \sigma(c_\alpha, \partial_{[\mu} c_{\nu]}, \partial, \Phi^*, R^{(1)}), \tag{5.8}$$

should be discarded since it is still a local reparametrisation, even though it is a non-trivial element of the $Q_0$-cohomology when restricted to the Hilbert space $\mathfrak{L}$. This latter property follows because

$$\delta_\Lambda^{(-1)}(\varphi) = \int d\varphi \, \delta_\Lambda^{(0)}(\varphi) \tag{5.9}$$

is not in $\mathfrak{L}_-$ (there is no choice of integration constant for which it is square integrable under (1.1)). In computing the cohomology, the same effect will arise more generally from discarding total derivative terms (with spacetime derivatives possibly applied multiple times). We see then that when dealing with cohomological descendants from the (interaction part of the) action $S$, we must allow in general for eigenoperators $\delta_\Lambda^{(n)}(\varphi)$ where $n$ is also a negative integer, defined formally through (1.2) or (4.11), and in practice by repeated $\varphi$ integrations. Nevertheless the interactions in $S$ itself must still lie in $\mathfrak{L}$ to preserve the Wilsonian RG structure [5,76], as explained at the end of sec. 4.1.1 and further discussed in the Conclusions.

## 6 Renormalized interactions to first order

The scaling dimension of the couplings $g_n^\sigma$, conjugate to the eigenoperators (4.28), is just given by their engineering dimension $[g_n^\sigma] = d - D_\sigma$ (as expected since we are expanding around the Gaussian fixed point). Although we could continue to keep $d$ general, we would do so from here on at the price of less clarity. So from here on we specialise to $d = 4$. Then

$$[g_n^\sigma] = 5 + n - d_\sigma. \tag{6.1}$$

As mentioned already in the Introduction, and discussed in ref. [5], in the continuum limit we must retain only the (marginally) relevant couplings $[g_n^\sigma] \geq 0$. The more field factors and/or derivatives included in $\sigma(\partial, \partial \varphi, h, \bar{c}, c, b, \Phi^*)$, the larger is its engineering dimension $d_\sigma$. But since we can take $n$ as large as we please, there is always an infinite tower of the eigenoperators (4.28) that are relevant. We are thus led to study the operator

$$f_\Lambda^\sigma(\varphi) \, \sigma(\partial, \partial \varphi, h, \bar{c}, c, b, \Phi^*) + \cdots, \tag{6.2}$$

where again we display only the top term, the tadpole corrections being determined by this and indicated by the ellipses. The couplings have been subsumed in a 'coefficient function'

$$f_\Lambda^\sigma(\varphi) = \sum_{n=n_\sigma}^\infty g_n^\sigma \delta_\Lambda^{(n)}(\varphi), \tag{6.3}$$

and $n_\sigma = 0$ if $d_\sigma \leq 5$, otherwise $n_\sigma = d_\sigma - 5$. Thus for $d_\sigma \geq 5$, we are including a marginal coupling $[g_{n_\sigma}^\sigma] = 0$. Since we will fully develop the theory only to first order in this paper, we treat it as though it is exactly marginal and include it. To decide whether it is actually marginally relevant (and thus kept) or marginally irrelevant (and thus discarded), we need to develop the theory to higher order in these couplings.

The diffeomorphism BRST invariance is successfully incorporated, if the renormalized interactions satisfy the appropriate Ward identity, which at first order is simply given by (2.52).

So far we have been discussing the bare interactions. We must now therefore derive the renormalized interactions.[11] Actually since we only consider the interactions to first order, the couplings $g_n^\sigma$ do not run and therefore at first sight there should be no difference in the structure from considering them to be bare or renormalized at this order. There is a difference however, because an infinite sum of terms (6.3) can have different properties from each term individually. This is in fact generically the case at sufficiently low scales $\Lambda$ [5].

From (4.2) and (4.5), the flow equation for $f_\Lambda^\sigma$ is

$$\dot{f}_\Lambda^\sigma(\varphi) = \tfrac{1}{2}\dot{\Omega}_\Lambda f_\Lambda^{\sigma\,\prime\prime}(\varphi). \tag{6.4}$$

First we note that we can regard $\Lambda$ not as the ultraviolet cutoff for the Wilsonian effective action, but as the infrared cutoff for a Legendre effective action [10, 67, 77], without any changes at this first order in the couplings. The physical operator

$$f^\sigma(\varphi)\,\sigma(\partial, \partial\varphi, h, \bar{c}, c, b, \Phi^*) \tag{6.5}$$

is then given by choosing a finite solution at finite $\Lambda$ (a.k.a. the renormalized operator) and removing the cutoff:

$$f^\sigma(\varphi) = \lim_{\Lambda \to 0} f_\Lambda^\sigma(\varphi). \tag{6.6}$$

Note that at this stage the tadpole corrections in (6.2), have disappeared, since they are all proportional to powers of $\Lambda$ (the regularised tadpole integrals).[12] Since (6.2) is a sum over eigenoperators, it is a solution of the flow equation provided that the $g_n^\sigma$ are independent of $\Lambda$. It is a finite solution provided that the $g_n^\sigma$ are finite and that certain convergence criteria are met. As we will review, these in turn determine properties of the physical operator.

For $\Lambda$ large, say $\Lambda = \Lambda_0$, the operator must be square integrable under (5.1). This is the quantisation condition. This means that $f_\Lambda^\sigma(\varphi)$ must be square integrable under (1.1). Using the orthonormality relations (4.12), we see that

$$\int_{-\infty}^{\infty} d\tilde{\varphi}\; e^{a^2\tilde{\varphi}^2}\left(\tilde{f}_\Lambda^\sigma(\tilde{\varphi})\right)^2 = \frac{a}{\sqrt{\pi}}\,\Lambda^{2d_\sigma-10}\sum_{n=n_\sigma}^{\infty} n!\left(g_n^\sigma\right)^2\left(\frac{2a^2}{\Lambda^2}\right)^n. \tag{6.7}$$

Since we require that the sum converges for $\Lambda = \Lambda_0$, we see that the sum will then converge for all $\Lambda \geq \Lambda_0$. On the other hand, the right hand side will have a radius of convergence determined by the $g_n^\sigma$, which we choose to label as the point $\Lambda = a\Lambda_\sigma$. Evidently, $0 \leq \Lambda_\sigma < \Lambda_0/a$ (with $\Lambda_\sigma = 0$ being an infinite radius of convergence). We call it the amplitude suppression scale, for reasons that will be clear in a moment.

For $\Lambda < a\Lambda_\sigma$ the coefficient function is no longer in $\mathfrak{L}_-$. As shown in ref. [5], there are two reasons why this can happen. Either the function $f_\Lambda^\sigma(\varphi)$ develops singularities, after which the flow towards the IR fails to exist, or it does so because $f_\Lambda^\sigma$ fails to decay fast enough at large $\varphi$. We need to choose the $g_n^\sigma$ so that the flow all the way to $\Lambda \to 0$ does exist. Then we know that asymptotically for large $\varphi$:

$$f_{a\Lambda_\sigma}^\sigma(\varphi) \sim \exp\left(-\frac{a^2\varphi^2}{2a^2\Lambda_\sigma^2}\right) = \exp\left(-\frac{\varphi^2}{2\Lambda_\sigma^2}\right). \tag{6.8}$$

The solution to the flow equation (6.4) can be written in terms a Fourier transform over the conjugate momentum $\pi$:

$$f_\Lambda^\sigma(\varphi) = \int_{-\infty}^{\infty}\frac{d\pi}{2\pi}\,\mathfrak{f}^\sigma(\pi)\,e^{-\frac{\pi^2}{2}\Omega_\Lambda + i\pi\varphi}, \tag{6.9}$$

---

[11]In the essay [6] we deliberately glossed over this point.

[12] As explained in ref. [5], they are there to cancel the remaining quantum corrections, thus resulting in an operator that is form invariant under lowering $\Lambda$.

where $\mathfrak{f}^\sigma$ is the Fourier transform of the physical $f^\sigma$, as is clear since $\Omega_\Lambda$ vanishes as $\Lambda \to 0$. From (6.8) and (6.9) we see that the large $\varphi$ behaviour at $\Lambda = a\Lambda_\sigma$, is reproduced by

$$\mathfrak{f}^\sigma(\pi) \sim e^{-\pi^2 \Lambda_\sigma^2 / 4}. \tag{6.10}$$

Setting $\Lambda = 0$ in (6.9), we thus see that the physical operator is characterised by the large $\varphi$ behaviour:

$$f^\sigma(\varphi) \sim e^{-\varphi^2 / \Lambda_\sigma^2}. \tag{6.11}$$

Taylor expanding $\mathfrak{f}^\sigma(\pi)$ in (6.9), and performing the $\pi$ integrals, reproduces the representation of $f_\Lambda^\sigma$ as a sum over eigenoperators (1.2). Comparing to (6.3), we thus see that

$$\mathfrak{f}^\sigma(\pi) = \sum_{n=n_\sigma}^{\infty} g_n^\sigma (i\pi)^n. \tag{6.12}$$

Since the $g_n^\sigma$ yield the series (6.7), which converges for $\Lambda > a\Lambda_\sigma$, we see that the above series has an infinite radius of convergence. Therefore $\mathfrak{f}^\sigma$ is an entire function.

To summarise, we have shown that the physical operator (6.5) is characterised by having a coefficient function $f^\sigma$ whose large amplitude behaviour (6.11) is exponentially suppressed by an amplitude suppression scale $0 \le \Lambda_\sigma < \Lambda_0/a$, and such that its Fourier transform $\mathfrak{f}^\sigma$ is an entire function, whose Taylor expansion (6.12) gives the couplings, and which has large $\pi$ behaviour given by (6.10). Using $\mathfrak{f}^\sigma$ we can then reconstruct the finite solution at finite $\Lambda$ from (6.9), and thus the renormalized operator (6.2).

## 6.1 Examples

It will be useful to consider some examples. Let $\sigma(\partial, \partial\varphi, h, \bar{c}, c, b, \Phi^*)$ be a dimension five operator. Dimension five is what we get for the $S_1$ operators obtained in standard quantisation of gravity, *cf.* sec. 7.1, which in turn leads to the associated coupling $[\kappa] = -1$ being irrelevant (non-renormalizable). In the new quantisation the monomials must be accompanied by the coefficient function (6.3) and for $d_\sigma = 5$, all the couplings are perturbatively renormalizable, with $g_0^\sigma$ being marginal and the rest ($g_{n>0}^\sigma$) relevant. For $f_\Lambda^\sigma(\varphi)$ we can lift the example given in ref. [5]. We set the physical coefficient function, (6.6), to

$$f^\sigma(\varphi) = \kappa\, e^{-\varphi^2 / \Lambda_\sigma^2}, \tag{6.13}$$

so that at $\varphi = 0$, one recovers Newton's coupling $\kappa$. Taking the Fourier transform, we read off from (6.12) that the odd-$n$ couplings vanish and the even-$n$ ones are given by ($m \ge 0$ integer):

$$g_{2m}^\sigma = \frac{\sqrt{\pi}}{m! 4^m} \kappa\, \Lambda_\sigma^{2m+1}. \tag{6.14}$$

Performing the integral in (6.9) gives the coefficient function at finite $\Lambda$:

$$f_\Lambda^\sigma(\varphi) = \frac{\kappa a \Lambda_\sigma}{\sqrt{\Lambda^2 + a^2 \Lambda_\sigma^2}} \exp\left(-\frac{a^2 \varphi^2}{\Lambda^2 + a^2 \Lambda_\sigma^2}\right). \tag{6.15}$$

We see explicitly that $f_\Lambda^\sigma(\varphi)$ exits $\mathfrak{L}_-$ as $\Lambda$ falls below $a\Lambda_\sigma$, through failure of the integral (6.7) to converge at large $\varphi$.

Recall that an undifferentiated $\varphi$ is excluded from $\sigma(\partial, \partial\varphi, h, \bar{c}, c, b, \Phi^*)$, the reason being that this $\varphi$ dependence must be described through the sum over the eigenoperators (6.3). At first sight we can relax this, and let $\sigma$ have undifferentiated $\varphi$, since such factors can be exchanged for eigenoperators using (5.2). But using this relation results in new couplings $g_n^\sigma$ that depend on $\Lambda$ through $\Omega_\Lambda = \Lambda^2/2a^2$, which then does not solve the flow equation (6.4)

since it requires that the $g_n^\sigma$ to be constant. Factors of $\varphi$ can be incorporated but one needs to put them in the physical coefficient function, and then work backwards using (6.9) to the coefficient function at finite $\Lambda$. (Recall that the natural RG flow for $\varphi$ is from the IR to the UV [5].) For example let us take

$$f^{\sigma'}(\varphi) = \kappa\,\varphi\,e^{-\varphi^2/\Lambda_{\sigma'}^2}, \tag{6.16}$$

where the remaining monomial $\sigma'$ now has dimension four. By considering the Fourier transform, or indeed formally from (5.2) at $\Lambda = 0$, we can relate this to the above example to see that the couplings are now

$$g_{2m+1}^{\sigma'} = -(2m+2)\,g_{2m+2}^{\sigma}|_{\sigma=\sigma'} = -\frac{1}{2}\frac{\sqrt{\pi}}{m!4^m}\kappa\,\Lambda_{\sigma'}^{2m+3} \qquad (m \geq 0), \tag{6.17}$$

(with the even index couplings vanishing). Evaluating (6.9) at finite $\Lambda$ gives:

$$f_\Lambda^{\sigma'}(\varphi) = \frac{\kappa\,a^3\Lambda_{\sigma'}^3}{\left(\Lambda^2 + a^2\Lambda_{\sigma'}^2\right)^{3/2}}\,\varphi\,\exp\left(-\frac{a^2\varphi^2}{\Lambda^2 + a^2\Lambda_{\sigma'}^2}\right). \tag{6.18}$$

Suppose now that $\sigma$ has dimension $d_\sigma = 6$. In this case the sum over eigenoperators (6.3) starts at $n_\sigma = 1$ so as to include only the marginal and relevant couplings, and exclude the irrelevant $[g_0^\sigma] = -1$. One way to construct a solution is to adapt the one above by subtracting the $n = 0$ piece. Thus (6.15) becomes (multiplying also by $\kappa$ to match dimensions, as would appear classically)

$$f_\Lambda^\sigma(\varphi) = \frac{\kappa^2 a\Lambda_\sigma}{\sqrt{\Lambda^2 + a^2\Lambda_\sigma^2}}\exp\left(-\frac{a^2\varphi^2}{\Lambda^2 + a^2\Lambda_\sigma^2}\right) - \kappa^2\Lambda_\sigma\sqrt{\pi}\,\delta_\Lambda^{(0)}(\varphi), \tag{6.19}$$

so that the physical coefficient function, and non-vanishing couplings, are

$$f^\sigma(\varphi) = \kappa^2\,e^{-\varphi^2/\Lambda_\sigma^2} - \kappa^2\Lambda_\sigma\sqrt{\pi}\,\delta(\varphi), \qquad g_{2m}^\sigma = \frac{\sqrt{\pi}}{m!4^m}\kappa^2\Lambda_\sigma^{2m+1} \qquad (m \geq 1). \tag{6.20}$$

This amounts to taking a linear combination of two coefficient functions, one with finite amplitude decay scale $\Lambda_\sigma$, and the other with vanishing amplitude decay scale. More interesting for our purposes is to keep both amplitude decay scales non-vanishing and thus choose:

$$f^\sigma(\varphi) = \frac{\kappa^2}{\gamma - 1}\left(\gamma\,e^{-\frac{\varphi^2}{\Lambda_\sigma^2}} - e^{-\frac{\varphi^2}{\Lambda_\sigma^2\gamma^2}}\right), \tag{6.21}$$

where $\gamma > 1$ (without loss of generality) is the ratio of the two amplitude decay scales, and we still normalise to $f^\sigma(0) = \kappa^2$. Then

$$f_\Lambda^\sigma(\varphi) = \frac{\gamma a\Lambda_\sigma\kappa^2}{\gamma - 1}\left[\frac{1}{\sqrt{\Lambda^2 + a^2\Lambda_\sigma^2}}\exp\left(-\frac{a^2\varphi^2}{\Lambda^2 + a^2\Lambda_\sigma^2}\right)\right.$$
$$\left. -\frac{1}{\sqrt{\Lambda^2 + a^2\gamma^2\Lambda_\sigma^2}}\exp\left(-\frac{a^2\varphi^2}{\Lambda^2 + a^2\gamma^2\Lambda_\sigma^2}\right)\right] \tag{6.22}$$

and the couplings are (integer $m \geq 0$)

$$g_{2m}^\sigma = \frac{\sqrt{\pi}}{m!4^m}\frac{\gamma}{\gamma - 1}(1 - \gamma^{2m})\kappa^2\Lambda_\sigma^{2m+1}, \tag{6.23}$$

where the ratio in (6.21) was chosen to ensure that $g_0^\sigma = 0$.

Finally consider a linear combination of such terms $f_\Lambda^\sigma = \sum_{k=1}^N a_k f_\Lambda^k$, each with their own amplitude decay scale $\gamma_k\Lambda_\sigma$. Clearly by appropriate choice of the $a_k$, we can set to zero in this way any finite number of couplings $g_n^\sigma$, while normalising to the natural scale $f^\sigma(0) = \kappa^{d_\sigma - 4}$.

# 7 BRST cohomology

## 7.1 Classical BRST cohomology

We briefly recall the standard case where the classical action $S_{cl}$ is developed perturbatively in $\kappa$, which then has the form of (2.32), after which quantum corrections are to be computed order by order in $\hbar$. We thus start with the Pauli-Fierz action for a single massless spin-2 field (3.1) and the free diffeomorphism algebra (3.2), and ask at the classical level how non-trivial interactions can be consistently incorporated into each of these. The apotheosis of these investigations was reached in ref. [22]. With some weak restrictions, it can be rigorously shown that there is only one solution, modulo field reparametrisations, namely the one implied by the Einstein-Hilbert action (3.3). Although that paper deals with multi-graviton theories, when restricted to the case of a single massless spin-2 field it recovers and somewhat generalises previous results [30–39].

The analysis proceeds in minimal gauge invariant basis. The first order perturbation $S_1$ satisfies (2.52). However since we work at the classical level, we set $\hbar = 0$ in the QME, which just amounts to setting the measure operator $\Delta = 0$, and thus work with the Classical Master Equation (2.13). As follows from (3.13), or (2.12), we thus have $s_0 = Q_0 + Q_0^-$, where from (3.16) we have

$$Q_0^2 = 0, \quad (Q_0^-)^2 = 0, \quad \{Q_0, Q_0^-\} = 0. \tag{7.1}$$

Grading $S_1$ by antighost number, $S_1 = \sum_n S_1^n$, we have

$$Q_0 S_1^n + Q_0^- S_1^{n+1} = 0, \tag{7.2}$$

analogous to (3.14).

The first assumption is that action functionals are local, in particular $S_1$ therefore has only a finite number of spacetime derivatives. When we say that $S_1$ is $s_0$-exact we mean that it can be written as $S_1 = s_0 K$, such that $K$ is also a local functional. It also means that for these purposes, we throw away boundary terms that are generated by integration by parts. Then it can be proven that all solutions to (7.2) for $n > 2$, are also cohomologically trivial in the space [21, 22]. By reparametrisation, the $S_1^{n>2}$ can therefore be set to zero. We are left with finding non-trivial solutions to[13]

$$Q_0 S_1^2 = 0, \quad Q_0 S_1^1 + Q_0^- S_1^2 = 0, \quad Q_0 S_1 + Q_0^- S_1^1 = 0. \tag{7.3}$$

Furthermore, it can be proven that in order to have a chance of satisfying the second equation, $S_1^2$ must have a single antifield and otherwise only ghost terms. Since these ghost terms must appear as in (5.7), ghost number and Lorentz invariance then determine $S_1^2$ uniquely

$$S_1^2 = \int d^4 x \, L_1^2, \quad \text{where} \quad L_1^2 = \partial_{[\alpha} c_{\beta]} c_\alpha c_\beta^*, \tag{7.4}$$

(the normalisation being absorbed into $\kappa$). Note that this is derived under the sole assumption that the action functionals are local (*i.e.* in particular have a finite number of derivatives) [22].

A major part of the power of this approach is that it handles infinitely many parametrisations simultaneously, in particular $S_1^2$ is defined only modulo the addition of $Q_0 K^2$. Any $K^2$ induced reparametrisation trivially solves the cohomology equations (7.3): we only need to add $Q_0^- K^2$ to $S_1^1$. By adding different exact pieces, for example we can treat $H_{\mu\nu}$ as contravariant in either or both indices, and/or as the first order in the expansion of the inverse

---

[13]Since $S^0$ has no fields with negative ghost number (*cf.* table 1) it coincides with the gauge invariant action $\mathcal{S}[H]$.

$g^{\mu\nu}$, similarly $c_\mu$ can be treated as contravariant or covariant, and with trivial changes, we can couch this in Cartan formulation. Beyond first order we can handle simultaneously an expansion of the fields in terms of any series in $H_{\mu\nu}$, the ghosts *etc.* For the present purposes a nice choice is to set $K^2 = -\frac{1}{2}H_{\alpha\beta}\,c_\alpha c_\beta^*$, and thus choose instead

$$L_1^2 = c_\alpha \partial_\beta c_\alpha\, c_\beta^* \,. \tag{7.5}$$

Operating with $Q_0^-$, and integrating by parts, the result is seen to be $Q_0$-exact, and thus

$$L_1^1 = 2c_\alpha \Gamma^{(1)\,\alpha}_{\ \ \beta\gamma} H_{\beta\gamma}^* \tag{7.6}$$

up to the addition of a piece that is non-trivial in the $Q_0$ cohomology. By (anti)ghost number, this latter piece must be linear in $H_{\mu\nu}^*$ and in $c_\alpha$ or $\partial_{[\mu}c_{\nu]}$. But then, by Lorentz invariance it must contain at least one curvature (3.11), since $\partial_{[\mu}c_{\nu]}H_{\mu\nu}^*$ vanishes. That makes the interaction contain at least three derivatives, which Boulanger *et al* exclude. Even if one allows for such interactions, the constraints provided by the last equation in (7.3), make it unlikely that there are any solutions [22].

Substituting (7.6) into the last equation in (7.3), and using (3.9), one finds a non-trivial solution which can be written as thirteen linearly independent terms. Up to of course integration by parts, this is $\mathcal{L}_1 = \mathcal{L}_{1\,cl}$, where

$$\begin{aligned}
\mathcal{L}_{1\,cl} =\ &2\varphi\partial_\beta H_{\beta\alpha}\partial_\alpha\varphi - 2\varphi(\partial_\alpha\varphi)^2 - 2H_{\alpha\beta}\partial_\gamma H_{\gamma\alpha}\partial_\beta\varphi + 2H_{\alpha\beta}\partial_\alpha\varphi\partial_\beta\varphi - 2H_{\beta\gamma}\partial_\gamma H_{\alpha\beta}\partial_\alpha\varphi \\
&+ \tfrac{1}{2}\varphi(\partial_\gamma H_{\alpha\beta})^2 - \tfrac{1}{2}H_{\gamma\delta}\partial_\gamma H_{\alpha\beta}\partial_\delta H_{\alpha\beta} - H_{\beta\mu}\partial_\gamma H_{\alpha\beta}\partial_\gamma H_{\alpha\mu} + 2H_{\mu\alpha}\partial_\gamma H_{\alpha\beta}\partial_\mu H_{\beta\gamma} \\
&+ H_{\beta\mu}\partial_\gamma H_{\alpha\beta}\partial_\alpha H_{\gamma\mu} - \varphi\partial_\gamma H_{\alpha\beta}\partial_\alpha H_{\gamma\beta} - H_{\alpha\beta}\partial_\gamma H_{\alpha\beta}\partial_\mu H_{\mu\gamma} + 2H_{\alpha\beta}\partial_\gamma H_{\alpha\beta}\partial_\gamma\varphi \,. \tag{7.7}
\end{aligned}$$

Again this solution is unique only up to addition of a piece that is non-trivial in the $Q_0$ cohomology. Since $\mathcal{S}_1$ can only depend on $H$, we mean equivalently that it is unique up to addition of terms invariant under linearised diffeomorphisms. If we insist that interactions with more than two derivatives are excluded, then there is only one new possibility:

$$\delta\mathcal{L}_1 = \lambda\varphi \,, \tag{7.8}$$

where $\lambda$ is a new coupling. We recognise that this is the linearised cosmological constant term. The invariant piece with two derivatives is a copy of the free graviton action (3.1), and thus can be absorbed by reparametrisation. If we relax the restriction on derivatives then powers of the curvatures (3.11) and their derivatives can be used, however this would take us down the route to higher derivative gravity, with its attendant problems [78].

To interpret what we have found, it is helpful momentarily to switch off the regularisation in (3.6). In fact we have no need of the regularisation here, since the analysis is and has been purely classical. Adding (7.6) to (3.6), and comparing to (2.2), we see that

$$(Q_0 + \kappa Q_1)H_{\mu\nu} = 2\nabla^{(1)}_{(\mu}c_{\nu)} \,, \tag{7.9}$$

where $\nabla^{(1)}_\mu$ is a covariant derivative, evaluated to first order in $\kappa$. If we regard $H_{\mu\nu}$ and $c_\nu$, as covariant tensors, it has the standard form. Thus we recognise that our choice of representatives of the BRST cohomology results in these assignments. We similarly read off from (7.5) that

$$(Q_0 + \kappa Q_1)c_\beta = -\kappa\,c_\alpha\partial_\beta c_\alpha \,. \tag{7.10}$$

The fact that $Qc_\beta$ is now non-vanishing, expresses the fact that diffeomorphisms do not commute beyond the linearised level. If $\kappa c^\mu$ is treated as a vector field,[14] this would just be half the Lie bracket:

$$Q'c^\mu = \frac{\kappa}{2}\mathfrak{L}_c c^\mu = \kappa\,c^\nu\partial_\nu c^\mu \,. \tag{7.11}$$

---

[14]See the comment on powers of $\kappa$ below (3.4).

From this we find, using (3.4):

$$Qc_\mu = Q'(c_\mu + \kappa H_{\mu\nu} c^\nu) = \kappa \left( c^\nu \partial_\nu c_\mu + 2\partial_{(\mu} c_{\nu)} c^\nu \right) + O(\kappa^2) = -\kappa \, c^\nu \partial_\mu c_\nu + O(\kappa^2), \qquad (7.12)$$

in agreement with (7.10). Finally (7.7) coincides with the triple graviton vertex one gets from the Einstein-Hilbert action (3.3), after expanding the metric using (3.4).

## 7.2 Quantum BRST cohomology

Locality is an important physical requirement, but it is also necessary for a non-trivial BRST cohomology. If we allow $K$ to be non-local then we can always write an $s_0$-closed $S_1$ as $S_1 = s_0 K$ [20,21], and a non-Abelian BRST algebra can be rewritten as an Abelian one [28]. (In the first case, an example can be found in [21] involving a further integral over time, and in second case one uses the ghost propagator.) It is therefore of the utmost importance to define the space of functionals over which the BRST cohomology is to be studied. In the usual framework, quantum corrections are inherently non-local. (Furthermore the measure operator $\Delta$ is usually ill-defined without further regularisation [23,29].) It is then unclear how to define a quantum BRST cohomology without further restriction.

As we have just reviewed, the standard procedure is to define it by requiring that it can be couched as a perturbation series in $\hbar$, starting from the classical case. In our formulation, we have a different route, one which is non-perturbative in $\hbar$ as we require. Locality is not abandoned but relaxed to quasi-locality (*cf.* footnote 2), provided by the effective action at finite cutoff $\Lambda$. Most importantly, as explained below (2.54) at first order the space of functionals is spanned by the eigen-operators (4.28) with constant coefficients (the couplings), and these eigenoperators are local. Phrasing the cohomology for renormalized operators, means we should instead use sums over (6.2), which again are local operators to be added together with constant coefficients. As we will see shortly, this is so constraining, it forces us to recover essentially the classical result.

Let us first assume the standard quantisation, where we take as eigenoperators polynomials in $\varphi$ also (ignoring the fact that these do not span the space of perturbations unless we rotate to imaginary $\varphi$ [5,76]). Then in our framework, the non-trivial solution (7.5) – (7.7) is not quite correct, because the triple graviton vertex is not an eigenoperator. It receives tadpole corrections from the flow equation (4.2), similar to that discussed in (4.27). Note that tadpole corrections are to be computed after first shifting to gauge fixed basis using (3.22), after which we map back to the gauge invariant basis (although in this case it is easy to see that these maps do not change the result). Then the only change is a graviton tadpole correction resulting in

$$\mathcal{L}_1 = \mathcal{L}_{1\,cl} + \frac{3}{2} b \Lambda^4 \varphi \,, \qquad (7.13)$$

where $b$ is the non-universal number [5]

$$b = \int \frac{d^4 \tilde{p}}{(2\pi)^4} \, C(\tilde{p}^2) \,. \qquad (7.14)$$

The correction is $Q_0$-closed as required, being a copy of (7.8). As per footnote 12, it is there to absorb the remaining quantum corrections. The physical operator is obtained in the $\Lambda \to 0$ limit, which sends us back to (7.7).

Our central thesis is that we should not be dealing with polynomial interactions for $\varphi$ but the (marginally) relevant eigenoperators that, together with the irrelevant ones, span a Hilbert space of interactions, as uniquely determined by the wrong-sign kinetic term and the Wilsonian RG. We want solutions to $s_0 S_1 = 0$ modulo the exact solutions $S_1 = s_0 K$, where $K$ itself is built from linear combinations of the eigenoperators with constant coefficients. Now from (3.13),

$s_0 = Q_0 + Q_0^- - \Delta^- - \Delta^=$, where the measure operators are defined in (3.12). We see that if the measure operators give a non-vanishing contribution, they do so by eliminating fields together with their space-time derivatives, replacing them with a regularised tadpole integral. Since we know that $s_0$ maps back into the space of eigen-operators with constant coefficients (*cf.* sec. 2.6), we see that the measure operators can only serve to reproduce the tadpole corrections that must have already been specified by the top terms (4.28) of the eigenoperators. These top terms are determined solely by the action of $Q_0 + Q_0^-$.

A simple example will illustrate the point. We use the formulae in sec. 3.1. Let

$$K_{ex} = K_{ex}^2 = -\int d^4x \; c^* \cdot c \, \delta_\Lambda^{(n)}(\varphi). \tag{7.15}$$

We will soon need to be more general, using a coefficient function (6.3), but for this illustration it is clearer to focus on just one of the terms, or equivalently specialising to the case where $g_n^\sigma = 1$ and all other couplings to zero. Mapping to gauge fixed basis via (3.22) has no effect, and no tadpole terms are generated by (4.2).[15] Therefore $K_{ex}$ is an eigenoperator as it stands (with dimension $2-n$). Operating with $s_0$, we get from (3.15) (using also (1.2), in this example $\Delta^-$ has no effect):

$$L_{ex}^2 = c^* \cdot c \, \partial \cdot c \, \delta_\Lambda^{(n+1)}(\varphi), \quad L_{ex}^1 = 2\partial_\mu H_{\mu\nu}^* c_\nu \, \delta_\Lambda^{(n)}(\varphi), \quad \text{and} \quad \mathcal{L}_{ex} = -4b\Lambda^4 \delta_\Lambda^{(n)}(\varphi), \tag{7.16}$$

where $b$ was defined above. Mapping these to gauge fixed basis, only changes $L_{ex}^1$, which becomes:

$$L_{ex}^1|_{\text{gf}} = 2\partial_\mu H_{\mu\nu}^* c_\nu \, \delta_\Lambda^{(n)}(\varphi) + \Box \bar{c}_\nu c_\nu \, \delta_\Lambda^{(n)}(\varphi). \tag{7.17}$$

We see that in this example the eigenoperator equation (2.54) exclusively generates tadpole corrections from this new piece. Computing it, we see that the eigenoperator is actually $L_{ex}^1|_{\text{gf}} + \mathcal{L}_{ex}$. Mapping back to gauge invariant basis, we see that $s_0$ maps the eigenoperator $K_{ex}^2$ to the sum of two eigenoperators with the same eigenvalue (engineering dimension), namely $L_{ex}^2$ and $L_{ex}^1 + \mathcal{L}_{ex}$. These eigenoperators are determined by their top terms, namely $L_{ex}^2$ and $L_{ex}^1$, and the top terms follow from the action of $Q_0$ and $Q_0^-$ alone.

Retaining only the top terms, we are left with $T^n \in S_1^n$ and to solve the cohomology of the equations[16]

$$Q_0 T^n + Q_0^- T^{n+1} = 0, \tag{7.18}$$

for all $n \geq 0$. These are of the form of the classical cohomology equations. We would now be able to apply ref. [22] directly, and recover the results of sec. 7.1, save for two points. Ref. [22] assumes that there are a finite number of derivatives, whereas the $T^n$ span a space that is only quasi-local and thus can have a derivatives to arbitrarily high order. On the other hand we know from sec. 6, that each monomial $\sigma$ in the $T^n$ must be accompanied by a coefficient function $f_\Lambda^\sigma(\varphi)$ that cannot be constant in $\varphi$ at finite overall cutoff, since its large field behaviour is constrained by the amplitude decay scale $\Lambda_\sigma < \Lambda_0/a$.

Since the $T^n$ are quasi-local, we can however split (grade) them according to the number of spacetime derivatives in each piece $T^n = \sum_{m=0} T_{\partial^m}^n$. If the action of $Q_0$ is non-vanishing it raises the number of derivatives by one by acting on $H_{\mu\nu}$, similarly $Q_0^- = Q_\partial^- + Q_{\partial^2}^-$ raises them by one via its action on $c^*$, or two via its action on $H^*$, respectively. Therefore we have:

$$Q_0 T_{\partial^m}^n + Q_\partial^- T_{\partial^m}^{n+1} + Q_{\partial^2}^- T_{\partial^{m-1}}^{n+1} = 0. \tag{7.19}$$

---

[15]Recall that the $\varphi$ piece of (4.2) is used in (4.7), equivalently (6.4), to derive the functional form of $\delta_\Lambda^{(n)}(\varphi)$.

[16]Apparently one could simply take the $\Lambda \to 0$ limit, forcing the vanishing of the measure operator, and then solving directly for the cohomology of the physical operators (6.5). However in this limit, the free action (3.6) diverges, the QME degenerates beyond first order *cf.* (2.53), and the action itself becomes non-local, *cf.* also comments in sec. 6.

Since the ghosts are Grassmann, $c_{\mu_1} c_{\mu_2} c_{\mu_3} c_{\mu_4} c_{\mu_5} = 0$, and thus the $T_{\partial^0}^{n \geq 5}$ necessarily vanish. Then (7.19) tells us that

$$Q_0 T_{\partial^0}^4 = 0. \tag{7.20}$$

Since

$$Q_0 f_\Lambda^\sigma(\varphi) = \partial \cdot c \, f_\Lambda^{\sigma\prime}(\varphi), \tag{7.21}$$

is non-vanishing, the only way for $T_{\partial^0}^4$ to solve (7.20) is for it to vanish. Then (7.19) tells us that $Q_0 T_{\partial^0}^3 = 0$. The same argument shows that $T_{\partial^0}^3$ must also therefore vanish. Iterating we thus show that $T_{\partial^0}^n$ vanishes for all $n$.

We will eliminate iteratively the higher derivative terms, in a closely similar way. We will need to characterise non-trivial solutions to $Q_0 T_{\partial^m}^n = 0$ for general $m$ and $n$. Extending the definition of $\mathfrak{L}$ for the descendents as in sec. 5, we can use directly the result of appendix A1 of ref. [22]: by discarding reparametrisations (trivial terms) any non-trivial $T_{\partial^m}^n$ can be made to satisfy this equation without integration by parts. This result relies on Theorem 3.1 of that paper: that the cohomology of the exterior derivative in the space of invariant polynomials, is trivial if the antifield number $n > 0$. However it is easy to see that the non-constant coefficient function $f_\Lambda^\sigma(\varphi)$ plays the same rôle here and that therefore for us Theorem 3.1 holds also for vanishing antifield number.

Now assume that we have eliminated all derivative terms, up to and including $\partial^{m-1}$. Since the $c_\mu$ are Grassmann, there is a maximum antighost number $T_{\partial^m}^n$. Since there are no $\partial^{m-1}$ pieces, (7.19) tells us that $Q_0 T_{\partial^m}^n = 0$. We just saw that a non-trivial solution can be taken to solve this without integration by parts. But such a solution must in its entirety be of the form (5.7). Since from sec. 6, each monomial must however have an $f_\Lambda^\sigma(\varphi)$ as a factor, which is not constant in $\varphi$, that is not possible unless in fact $T_{\partial^m}^n$ vanishes. Now (7.19) tells us that $Q_0 T_{\partial^m}^{n-1} = 0$. Thus by iteration we establish that there is no non-trivial $T_{\partial^m}^n$ for all $n$, and thus by iteration this also true for all $m$. We have therefore shown that there is only a trivial quantum BRST cohomology so long as each monomial $\sigma$ has a non-constant $f_\Lambda^\sigma(\varphi)$ as a factor.

## 7.3 Recovering diffeomorphism invariance

Therefore the only way we can recover a non-trivial quantum BRST cohomology, and thus incorporate beyond the free level a sensible notion of diffeomorphism invariance, is for the $f_\Lambda^\sigma(\varphi)$ to become independent of $\varphi$. Fortunately this is in fact possible for the renormalised coefficient functions, by taking the limit in which all amplitude decay scales are sent to infinity.

Recall that the amplitude decay scales must satisfy $0 \leq \Lambda_\sigma < \Lambda_0/a$. Therefore the $\Lambda_\sigma$ must start out finite. Indeed as we saw in sec. 6, the amplitude decay scale is set by the renormalized couplings which themselves must be finite. In constructing the theory we must include all the (marginally) relevant bare couplings at values induced by requiring finite couplings at physical scales. We then send $\Lambda_0 \to \infty$ to form the continuum limit. At this point we have a perturbatively renormalizable interacting theory, although with an infinite number of couplings $g_n^\sigma$ and no diffeomorphism invariance.

We can now however send the $\Lambda_\sigma \to \infty$. For $S_1$ for example, we can take the positive antifield number pieces (7.5) and (7.6), and multiply them by coefficient functions of the form (6.15). Then it is easy to see that they are valid renormalized operators in the new quantisation, i.e. of form (6.2), in particular without tadpole corrections. For any finite $\Lambda$ and finite $\varphi$, we have

$$\lim_{\Lambda_\sigma \to \infty} f_\Lambda^\sigma(\varphi)/\kappa \to 1. \tag{7.22}$$

Thus in this limit $S_1^2$ and $S_1^1$ become the standard non-trivial solutions for the first two equations in (7.3). For $\mathcal{S}_1$ we have to use the different form (6.18) of coefficient function for the pieces which contain an undifferentiated $\varphi$, and we also have to recognise that it is not yet a linear

combination of eigenoperators. Extracting the undifferentiated $\varphi$ pieces from (7.7) by using (3.28), we thus find the full operator is made up of twelve top terms and a tadpole contribution:

$$
\begin{aligned}
\mathcal{L}_1 = \Big(&\frac{1}{4}h_{\alpha\beta}\partial_\alpha\varphi\partial_\beta\varphi - h_{\alpha\beta}\partial_\gamma h_{\gamma\alpha}\partial_\beta\varphi - \frac{1}{2}h_{\gamma\delta}\partial_\gamma h_{\alpha\beta}\partial_\delta h_{\alpha\beta} - h_{\beta\mu}\partial_\gamma h_{\alpha\beta}\partial_\gamma h_{\alpha\mu} \\
&+ 2h_{\mu\alpha}\partial_\gamma h_{\alpha\beta}\partial_\mu h_{\beta\gamma} + h_{\beta\mu}\partial_\gamma h_{\alpha\beta}\partial_\alpha h_{\gamma\mu} - h_{\alpha\beta}\partial_\gamma h_{\alpha\beta}\partial_\mu h_{\mu\gamma} + \frac{1}{2}h_{\alpha\beta}\partial_\gamma h_{\alpha\beta}\partial_\gamma\varphi\Big)f_\Lambda^\sigma(\varphi) \\
&+ \Big(\frac{3}{8}(\partial_\alpha\varphi)^2 - \frac{1}{2}\partial_\beta h_{\beta\alpha}\partial_\alpha\varphi - \frac{1}{4}(\partial_\gamma h_{\alpha\beta})^2 + \frac{1}{2}\partial_\gamma h_{\alpha\beta}\partial_\alpha h_{\gamma\beta}\Big)f_\Lambda^{\sigma'}(\varphi) + \frac{3}{2}b\Lambda^4 f_\Lambda^{\sigma'}(\varphi). \quad (7.23)
\end{aligned}
$$

The tadpole contribution arises in the same way as it did in the standard quantisation, *cf.* the beginning of sec. 7.2. Indeed it is easy to see that the first eight terms above cannot contribute because their tadpoles are proportional to $h_{\alpha\alpha} = 0$, while the three of the remaining four terms that do contribute, give precisely the same contribution as they did in (7.13). In here we could have chosen to use a different $f_\Lambda^\sigma(\varphi)$ for each of the first eight terms, and similarly a different $f_\Lambda^{\sigma'}(\varphi)$ for each of the last four terms. In future when we consider quantum corrections at $O(\kappa^2)$ and higher, we will have to (see also sec. 8). The main point is that since

$$
\lim_{\Lambda_{\sigma'}\to\infty} f_\Lambda^{\sigma'}(\varphi)/\kappa \to \varphi, \quad (7.24)
$$

in the limit of infinite amplitude decay scales (7.22) and (7.24), we recover also the standard non-trivial quantum form for $\mathcal{S}_1$ and thus the full non-trivial solution to (7.3).

Clearly then we can recover the standard non-trivial BRST cohomology, for any of the parametrisations discussed in sec. 7.1. Although the coefficient functions do not diverge in this limit, for fixed $\kappa$, the couplings themselves (6.14), (6.17), do diverge in this limit. However we can treat them perturbatively. At one level this is akin to the usual artifice of perturbative renormalization by subtracting divergent counterterms. We can ensure that the couplings do remain small however, by requiring $\kappa$ to vanish faster than any power of $\Lambda_\sigma$ (for example as $\kappa \propto e^{-\Lambda_\sigma/\mu}$), forming ratios such as in (7.22) to extract the coefficients of the perturbative series in $\kappa$. (As in standard cases, we would expect this series to converge only in the asymptotic sense.)

For $S_1$ this is the complete analysis. The infinite number of couplings $\{g_n^\sigma, g_n^{\sigma'}\}$ that we have at the bare level, get traded for the single coupling $\kappa$ at the renormalized level, and we recover exactly the standard description.

The higher order parts $S_n$ need to be treated beyond the linearised level used in this paper, but let us put that aside for the moment to understand what we can so far in these cases. At the linearised level the bare couplings are the same as the physical ($\Lambda = 0$) couplings. The latter can be extracted from the physical coefficient function by computing [5]

$$
g_n^\sigma = \frac{(-)^n}{n!} \int_{-\infty}^{\infty} d\varphi\, \varphi^n f^\sigma(\varphi). \quad (7.25)
$$

The $S_n$ would have successively higher dimension monomials $\sigma$. For example local terms in $S_2$ will have monomials of dimension $d_\sigma = 6$. Then their $g_0^\sigma$ is irrelevant and in the continuum limit, needs to vanish at the bare level. From (7.25) that would require having a coefficient function whose integral vanishes. It is still possible to have such a thing and have it tend to a constant pointwise for finite $\varphi$ and $\Lambda$. Indeed we provided an example in (6.21) – (6.23). It is straightforward to see that (6.21) integrated over $\varphi$, does indeed vanish. This coefficient function has the characteristic that for large $\Lambda_\sigma$ and finite $\varphi$ and $\gamma$, it satisfies $f^\sigma(\varphi) = \kappa^2$ to exponential accuracy. But for $\varphi \gtrsim \Lambda_\sigma$ it gently changes sign and then decays away exponentially.

We saw in general in sec. 6 that for some monomial $\sigma$, it is possible to zero any finite number of couplings by choosing linear combinations, where clearly again as $\Lambda_\sigma \to \infty$ the coefficient function tends to a constant ($\kappa^{d_\sigma-4}$ in this case). More generally, we see from (7.25) that we need only choose a coefficient function such that $f^\sigma(\varphi)$ tends to a constant pointwise in $\varphi$, but such that the corresponding moments vanish, this vanishing being enabled by behaviour at large (and eventually infinite) $\varphi$.

## 8 Discussion

The above observations serve to emphasise some important points. The (bare) couplings are not directly related to the physical interactions, in a sense which is much more extreme than in a normal quantum field theory. Even at the linearised level where there is no difference between bare and physical couplings, the connection to the physical interactions is rather indirect. We see from (7.25) that the effect of each coupling is distributed non-locally through the (physical) coefficient function, so that its influence cannot be determined by considering only finite $\varphi$. Recovering diffeomorphism invariance involves infinitely many couplings in $S_1$ being traded at the renormalized level for a single effective coupling $\kappa$. At higher orders they would be traded for an effective coupling proportional to appropriate powers of $\kappa$. In this process, any finite number of renormalized $g_n^\sigma$ can set to zero without changing this effective coupling.

Therefore it seems meaningless to try to count the number of independent bare couplings. Instead we should ask how many free parameters remain in the renormalized theory. This has to be one of the next most important questions to answer. However to do so requires going beyond the linearised level to the higher order parts $S_n$. Then we will encounter new issues, some of which were already touched on in ref. [5].

We should make one comment on the phenomenology. Since recovering diffeomorphism invariance involves sending amplitude decay scales $\Lambda_\sigma \to \infty$, it would seem to rule out all but infinite inhomogeneity protection effects ("cosmic censorship") [5,8], although since this involves a limit in which also divergences are involved it may be that qualitatively similar physical effects survive, for example through logarithmic effects.

By appropriate choice of coefficient function, we can make a renormalizable interaction out of any monomial $\sigma$ no matter how large its dimension. For example we can use (6.22) for

$$\left(R_{\mu\nu}^{(1)}\right)^2 f_\Lambda^\sigma(\varphi), \tag{8.1}$$

since $d_\sigma = [\sigma] = 6$ and thus the non-renormalizable coupling $g_0^\sigma$ must vanish, as discussed at the end of sec. 7.3. Since we are dealing with a perturbatively renormalizable theory, we know that we never have to introduce this bare $g_0^\sigma$, and therefore it can remain zero to all orders in perturbation theory. The monomial in (8.1) is one of the dangerous operators that has to be introduced in the standard quantisation. If it were to be given a separate existence at the bare level, then it would be part of higher derivative gravity [78], allowing the theory to be perturbatively renormalizable but at the expense of unitarity (in flat space). The operator (8.1) does not challenge unitarity here, because there must also be the non-trivial function $f_\Lambda^\sigma(\varphi)$, and therefore it is not a contribution to the bilinear kinetic term for $H_{\mu\nu}$. If the operator (8.1) has to be introduced there is no reason to expect $f_\Lambda^\sigma$ to tend to a constant at the bare level (see also below). In fact, until we go beyond linearised order, we do not know whether we will be forced to include (8.1) and similar terms at the bare level. Given the remarks above, nor do we know what influence they would leave on the renormalized action, even if they have to be introduced. Fortunately already a computation at $O(\kappa^2)$ will clarify these issues.

Once we go to higher order, the couplings in $S_1$ will no longer be constants, but will run with the cutoff scale: $g_n^\sigma = g_n^\sigma(\Lambda)$ [5]. Indeed by inspection of (7.23) and considering the form of the $O(\kappa^2)$ quantum corrections that can be made out of these, it can be seen that each coefficient function will already run at $O(\kappa^2)$ and differently for each monomial. We see that we need to be careful to distinguish between renormalized couplings associated to different monomials, and also between these and the corresponding couplings at the bare scale $\Lambda_0$. In particular it is the bare scale irrelevant couplings that need to be sent to zero in the continuum limit $\Lambda_0 \to \infty$.

The irrelevant couplings at physical scales will then not vanish but take on values determined by the rest of the theory, through satisfying the flow equation (2.55) in tandem with the QME (2.4). In order to compute these pieces, we expand the action $S$ in $\kappa$ as in (2.32), where these powers of $\kappa$ are now to be viewed as an overall factor in the couplings $g_n^\sigma$, as in the examples in sec. 6.1.

Unlike in the standard quantisation, diffeomorphism invariance at the non-linear level, does not exist independently of the quantum corrections, but arises simultaneously, as envisioned in ref. [5]. We have seen in secs. 7.2 and 7.3, that to $O(\kappa)$ it must be the same as the classical realisation (sec. 7.1). But we do not know what the $O(\kappa^{n \geq 2})$ pieces will look like until the $S_{n \geq 2}$ are determined. These have to satisfy the consistency equations (2.53) (although at this stage it will be better rephrase all this in terms of the infrared cutoff Legendre effective action $\Gamma_\Lambda$, so that the physical correlators can be accessed directly [5]). As noted at the beginning of sec. 7.3, we have to take the limits in a prescribed order. We have to construct the continuum theory first, where the QME will not be satisfied, and then take a limit to a point where the QME will be satisfied at least as $\Lambda \to 0$. We have seen at linearised order that this can be done for any gauge invariant operator, without violating renormalizability. Even though this shows that there is enough parametric degrees of freedom beyond linearised order, we do not know yet whether in practice one can satisfy the QME using this freedom. For example one has to face the fact that as a consequence of (5.3), the flow of the product of two non-trivial coefficient functions is not given by multiplying the two $f_\Lambda^\sigma(\varphi)$ together, even at the linearised level [5]. Again how these effects are incorporated into the QME, will be clarified by the $O(\kappa^2)$ computation.

All of these effects provide reasons to expect a quantum gravity with different phenomenological properties compared to that obtained by the usual route. Clearly however, there are still many issues to be understood, many of which we expect will be clarified at $O(\kappa^2)$.

# 9   Conclusions

There are still many questions to be answered, nevertheless this structure offers a new way of quantising gravity. What we find most persuasive about it, is that the structure is not introduced *Deus ex machina* but is demanded by the theory itself if one insists on applying the Wilsonian RG directly to General Relativity without further modification. Most significant is that in this way we realise quantum gravity as a genuine continuum quantum field theory. It is well appreciated that the Wilsonian RG provides the framework to construct continuum quantum field theory. The starting point is flat Euclidean $\mathbb{R}^4$, an ultraviolet fixed point, and the infinite set of eigenoperator perturbations about this. What seems less well appreciated, is that in order for the Wilsonian RG to make sense, already at this level, it must be possible to write arbitrary linear perturbations within some suitably defined space, as a convergent sum over these eigenoperators. (For counter examples see refs. [5, 76].) This basic requirement is needed so that the flow of this perturbation then follows from the flow of the underlying couplings, as determined for example by their relevancy or irrelevancy. Such a structure does

not need to be imposed from outside. It is already provided by theory. The eigenoperator equations for the ghosts $c_\mu$ and the fluctuation fields $h_{\mu\nu}$ and $\varphi$, are of Sturm-Liouville type. The resulting space, $\mathfrak{L}$, is the Hilbert space defined by these Sturm-Liouville measures.

It so happens that in normal quantum field theories (ones with the right sign kinetic term) this property tells us that around the Gaussian fixed point (free field theory), the polynomial interactions that are widely used, do have the correct convergence properties, forming the Hilbert space $\mathfrak{L}_+$. But for the conformal factor part of the metric, $\varphi$, as a consequence of its wrong sign kinetic term, the space of such bare interactions has to be $\mathfrak{L}_-$: one in which the interactions are exponentially damped at large amplitude, more precisely, square integrable under (1.1).

This is the crucial condition that defines the new quantisation, one that is moreover preserved by the quantum corrections, as we verified in sec. 5. The full measure is the one given in (5.1), and the eigenoperators that span the Hilbert space are those in (4.28). Diffeomorphism invariance tells us that already at the classical level, there are interactions of arbitrarily high power in the fluctuation field. In the usual perturbative quantisation these are all irrelevant, *i.e.* non-renormalizable, and it is this property that forbids the construction of a non-trivial continuum limit about the Gaussian fixed point.[17] This should be contrasted with the new quantisation where infinitely many of the eigenoperators are relevant. As we have seen, in the new quantisation, by taking a limit to the boundary of $\mathfrak{L}$, these interactions can then be constructed at the linearised level, in such a way that they contain only (marginally) relevant operators, *i.e.* such that the theory remains renormalizable.

The Hilbert space structure $\mathfrak{L}_-$, and the corresponding operators $\delta_\Lambda^{(n)}(\varphi)$, were discovered in ref. [5]. The consequences of this structure, and also $\mathfrak{L}_+$, were developed logically step by step, in ref. [5], see also ref. [8]. But the application to gravity was treated only heuristically: in particular it was treated at the classical level where the problem can be reduced to parametrisation of the metric $g_{\mu\nu}$. The eigenoperators (4.28) are however non-perturbative in $\hbar$ so, as acknowledged in ref. [5], such a classical limit does not exist. Instead the theory must be constructed non-perturbatively in $\hbar$. Since it is constructed around the Gaussian fixed point, one can however develop it perturbatively in $\kappa$.

In this paper we have completed some major goals in this development, again proceeding step by step to develop the structure that is inherent in the theory. A crucial question left unanswered in ref. [5] was how diffeomorphism invariance is to be incorporated. In this paper we have fully answered that question to first order in $\kappa$. This requires constructing a well-defined action of the quantum BRST transformations (the QME), in a way that is consistent with the Wilsonian RG structure, the eigenoperators and $\mathfrak{L}$. A priori, this structure could have allowed a very different realisation of the quantum BRST cohomology, from the one implied by the normal quantisation. However we found in sec. 7.2 that only a trivial quantum BRST cohomology exists while remaining strictly inside $\mathfrak{L}$. To recover a non-trivial BRST cohomology, one must take a limit so that the renormalized coefficient functions become independent of $\varphi$. In sec. 7.3, we showed that this follows from diverging amplitude decay scale $\Lambda_\sigma$, a limit which is allowed after first taking the continuum limit $\Lambda_0 \to \infty$. In this way, the standard non-trivial BRST cohomology for diffeomorphism invariance is recovered without disturbing the renormalizability properties.

Thus the construction of perturbatively renormalizable quantum gravity is achieved only by breaking the gauge invariance at intermediate stages, not only through the use of a cutoff that makes visible the quadratic divergences which are central to the definition of the theory through (1.1), but through the fact that this leads to non-trivial coefficient functions $f_\Lambda^\sigma(\varphi)$ which are required to be present in every interaction, in order to define the continuum limit. Only after this is done, can we finally remove this structure at the renormalized level and

---

[17]One would also have to analytically continue the $\varphi$ integral [7] in particular to recover convergence [76].

regain diffeomorphism invariance.

The first step that has to be taken in order to derive the results above, is to combine the Wilsonian RG and the QME in such a way that a well-defined quantum BRST cohomology results, namely one for which the full free BRST charge $s_0$ (in particular the measure operator $\Delta$) is well defined when acting on arbitrary local functionals, and furthermore such that the cohomology can be restricted to the space spanned by the eigenoperators with constant couplings. This synthesis was achieved in sec. 2.4, in particular if $K$ is a linear combination of eigenoperators with constant couplings, then the cohomologically trivial operator $s_0 K$ is also a linear combination of eigenoperators with constant coefficients (see sec. 2.6). In sec. 5, we extended this definition so that cohomologically trivial interactions coincide with (BRST closed) physically trivial interactions (*i.e.* ones that satisfy the QME but are just generated by quasi-local reparametrisations of the free theory). As we saw in sec. 2.7, to study the quantum BRST cohomology, we are free to use the minimal gauge invariant basis, where the results are clearly independent of gauge fixing choices.

Specialising to quantum gravity, we saw that the quantum BRST cohomology can be graded by antighost number, leading almost immediately to the nine anticommutation identities (3.16), where furthermore at the free level one has the non-vanishing anticommutation relations (3.19) or (3.26) in gauge invariant or gauge fixed basis respectively.

On the other hand the eigenoperators themselves are defined in the gauge fixed basis. These are derived in sec. 4, culminating in the expression (4.28) for a general eigenoperator. At the same time, this yields the Hilbert space $\mathfrak{L}$ of bare interactions, as discussed in sec. 5. The BRST cohomology we are interested in is however the one that pertains to the renormalized interactions. As explained in sec. 6, developing further ref. [5], the renormalized interactions have different properties. This is so, even though we are working at the linearised level, so that the couplings $g_n^\sigma$ do not run with scale. For each monomial $\sigma$, we have a tower of relevant operators together with their couplings, which thus form a coefficient function $f_\Lambda^\sigma(\varphi)$. While the series defining $f_\Lambda^\sigma(\varphi)$ converges at the bare level, it typically fails to converge at some finite scale. This leads to the emergence of an amplitude decay scale $\Lambda_\sigma$. These properties are further explored in particular in sec. 6.1. The general form of the renormalized operator is given by (6.2) and the BRST cohomology is to be defined on the space which is a linear combination of such terms, suitably extended as described in sec. 5.

In sec. 7.2, we turn finally to the quantum BRST cohomology. A crucial observation is that the measure operators $\Delta^-$ and $\Delta^=$ can only contribute terms proportional to powers of $\Lambda$, and thus can contribute only to the tadpole corrections and not to the top terms in (6.2). The top terms must thus satisfy a free BRST cohomology of their own, (7.18), involving only the BRST charge $Q_0$ and the Kozsul-Tate differential $Q_0^-$. Then by grading these top terms by the number of spacetime derivatives, we can borrow from refs. [21, 22] to show that only a trivial quantum BRST cohomology exists so long as each monomial has a factor $f_\Lambda^\sigma(\varphi)$ that is non-constant in $\varphi$.

As already reviewed, we show how the limit of diverging amplitude decay scale allows us to remove this last restriction whilst remaining renormalizable, and thus recover the standard non-trivial BRST cohomology, *i.e.* correctly incorporate diffeomorphism invariance at first order in $\kappa$, furthermore in a way that reproduces standard realisations. In secs. 7.3 and 8 we highlighted that any diffeomorphism invariant operator can in this way be recovered at first order in $\kappa$, in particular no matter how high the dimension $d_\sigma$ of the monomial. Continuing to work at first order, we also saw that the map from the couplings $g_n^\sigma$ to the effective coupling $\kappa$ is not unique. In particular any finite number of $g_n^\sigma$ can be set to zero without changing this effective coupling. Many open questions remain, in particular it is not yet clear how many free parameters are required in the renormalized theory. However as explained in sec. 8, we expect the structure of the theory to become much clearer when perturbative corrections at

$O(\kappa^2)$ are computed.

## Acknowledgments

I acknowledge support from both the Leverhulme Trust and the Royal Society as a Royal Society Leverhulme Trust Senior Research Fellow, and from STFC through Consolidated Grants ST/L000296/1 and ST/P000711/1.

## A  More details on combining the QME and Wilsonian RG

The antibracket and measure operator satisfy many nice identities, some of which can be found in refs. [26,27,29]. Here we list some that we use. It is important to recognise that these identities hold true with the regularisation in sec. 2.6. We also give some identities that follow from linearised BRST invariance, and some further details on the canonical transformations encountered in this paper.

For functionals $X$ and $Y$, we use the symmetry property

$$(X,Y) = -(-)^{(X+1)(Y+1)}(Y,X),\tag{A.1}$$

in deriving (2.9). By a functional $X$ in the exponent we mean 0 (1) if $X$ is bosonic (fermionic). The nilpotency relation (2.9) follows from ($Z$ another functional)

$$(X,YZ) = (X,Y)Z + (-)^{Y(X+1)}Y(X,Z),\tag{A.2}$$

$\Delta^2 = 0$ and

$$\Delta(X,Y) = (\Delta X,Y) - (-)^X(X,\Delta Y).\tag{A.3}$$

Inspecting the above equations, it is useful to note that the functionals $X$, $Y$, *etc.* behave with opposite grading when they are inside antibrackets. To derive (2.36), we use

$$\Delta(XY) = (\Delta X)Y + (-)^X X\Delta Y + (-)^X(X,Y).\tag{A.4}$$

Note that this is different from the equation quoted in ref. [29] because we defined $\Delta$ in (2.5) via left-derivatives. Differentiating the antibracket with respect to a field $Z$, we have for example

$$\frac{\partial_r}{\partial Z}(X,Y) = \left(X,\frac{\partial_r Y}{\partial Z}\right) + (-)^{Z(1+Y)}\left(\frac{\partial_r X}{\partial Z},Y\right),\tag{A.5}$$

which in particular implies

$$\frac{\partial_r}{\partial\Phi^B}(S,S) = 2\left(\frac{\partial_r S}{\partial\Phi^B},S\right).\tag{A.6}$$

Note that the (linearised) BRST invariance of $\mathcal{S}_0$ implies by (2.21) and (2.27):

$$R^D_{\ B}\,\triangle^{-1}_{DE} + R^D_{\ E}\,\triangle^{-1}_{DB} = 0\tag{A.7}$$

where in symmetrising we note that $B$ and $E$ have the opposite statistics. We also note that these symmetry statements are unaffected by multiplying through by cutoff functions, so we drop the cutoffs for now. In gauge fixed basis, we can multiply through by $\triangle^{BA}\triangle^{EC}$,and thus the free propagators satisfy

$$R^C_{\ B}\,\triangle^{BA} + R^A_{\ B}\,\triangle^{BC} = 0\,.\tag{A.8}$$

These BRST invariance relations can be cast as symmetry relations. For example transposing the propagators in (A.8) by using (2.24), and noting again that the free indices have opposite statistics

$$\triangle^{AB} R^C_{\ B} = \triangle^{CB} R^A_{\ B} \,. \tag{A.9}$$

We see that $\Psi^*$ in eqn. (2.47) is already written in symmetric form. We also note that it is overall fermionic as it should be.

For $K$ in (2.46) to be a finite quantum canonical transformation, it must leave the QMF (2.4) invariant. Since it is written in the form of a finite classical canonical transformation, for the antibracket this is already clear [28]. Since it implies that

$$\check{\Phi}^*_A = \Phi^*_A, \qquad \text{and} \qquad \check{\Phi}^A = \Phi^A + \frac{\partial_l}{\partial \Phi^*_A} \Psi^*[\Phi^*], \tag{A.10}$$

the measure operator is also invariant because (where $\check{\Delta}$ is the operator for transformed fields):

$$\Delta S = \check{\Delta} S + \frac{\partial_l^2 \Psi^*}{\partial \Phi^*_A \partial \Phi^*_B} \frac{\partial_l^2 S}{\partial \check{\Phi}^A \partial \check{\Phi}^B} \tag{A.11}$$

and the last term vanishes by the opposite statistics of $\check{\Phi}^A$ and $\Phi^*_A$.

Alternatively one can start from the general form of a finite quantum canonical transformation which takes the general form of a finite classical canonical transformation, *i.e.* (2.46), but also takes into account the Berezinian $J$, the super-Jacobian of the change to new variables [28, 29, 79]:

$$S[\check{\Phi}, \check{\Phi}^*] = S[\Phi, \Phi^*] - \tfrac{1}{2} \ln J \,. \tag{A.12}$$

Then it is straightforward to see that for (A.10), $J = 1$, and thus the canonical transformation in this case has vanishing quantum part.

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
