# Peer review of "Quantum gravity, renormalizability and diffeomorphism invariance"

_SciPost Physics, doi:SciPost Phys. 5, 040 (2018)_

## Round 1 · Referee Report · Anonymous (Referee 1) · 2018-9-6

Strengths

1- The article is clearly written and scientifically correct. It builds on concepts that are well-established within the framework of quantum field theory. 2- The article discusses the consequences of applying the Wilsonian Renormalization Group to General Relativity. The novel viewpoint taken in this approach may lead to new insights on the quantization of gravity which are complementary to the ones found in other quantum gravity programs.

Weaknesses

1- Owed to the topic discussed, the work is rather technical. Thus the article is mostly of interest to experts working on the specific field.

Report

The work focuses on the perturbative quantization of gravity based on the free field fixed point. It continues the investigation initiated in arXiv:1802.04281 [hep-th] and arXiv:1804.03834 [hep-th] by elucidating the structure of the underlying Hilbert space and performing a detailed analysis of the BRST cohomology associated with diffeomorphism invariance. It is shown that Newton’s constant and standard diffeomorphism invariance can be recovered by going to the boundary of the Hilbert space.

An intriguing observation made in this series of works is that applying the Wilsonian renormalization group directly to general relativity leads to a the Hilbert space which is “exotic” in the sense that eigenperturbations around the free theory include fluctuation fields to infinitely high powers. Given the starting point of the construction, general relativity, one may wonder if there is a relation to the Hilbert spaces studied in the context of Loop Quantum Gravity. It would be interesting to learn the author's thoughts in this direction.

Despite its technical nature the work is clearly written and all steps can be followed in detail. While it does not clarify all the questions related to the program, it clearly constitutes a major step forward which warrants publication in its own right. Thus I recommend the article for publication, leaving it to the author to decide whether the optional comments should be included in the final version.

Requested changes

1- In order to make the overall discussion more accessible, the author may include a table summarizing the notation introduced, e.g., in section 2. It is left to the author to decide if this suggestion is implemented in the published version of the article.

---

## Round 1 · Referee Report · Anonymous (Referee 2) · 2018-10-22

Strengths

1- novelty 2- rigor

Weaknesses

1-not clear that this approach will work. (But one could say the same of all current approaches to quantum gravity, except for the effective field theory approach, that demonstrably works within its domain of validity.)

Report

This paper develops technical aspects of a highly original research line recently initiated by the author. Searching for nonperturbative definitions of the UV limit for quantum gravity, he realized (ref.[5]) that it may be possible to construct a limit that is perturbative in Newton's constant, but non-perturbative in Planck's constant. Quite intriguingly, the key to the new approach lies in what is usually regarded as a nuisance, namely the indefiniteness of the Euclidean action for the conformal factor. The author claims that a consistent quantization of the theory can be constructed, provided we focus on the right set of operators, namely functions of the fields that are integrable in the natural inner product. These operators involve exponentials of the conformal factor and are very far from the polynomials that are used in conventional approaches.
The main step forward of this paper is a detailed analysis of the BRST cohomology at first order in Newton's coupling, given in section 7. It is clearly explained how this analysis differs from the standard one.
It is still doubtful that this approach will be ultimately successful. However, it provides a remarkable new point of view on various aspects of quantum gravity. I therefore recommend very strongly that it be accepted for publication.

Requested changes

spelling of DeWitt on page 6.

---

## Editorial Decision

published